# Green synthesis of silver nanoparticles from *Brownlowia tersa* leaf extract: multifaceted evaluation of antibacterial, antioxidant, cytotoxic, and anti-alzheimer potential

Md Ridoy Hossain[1], Md Al Saber[1], Md. Anisul Hoque[1], Md. Shamsur Rahman[2], Florence Bornali Ratno[1], Md. Nazmul Hasan Zilani[2], Md. Ohiduzzaman[3], Md. Nazmul Hasan ![ORCID][1]*

**1** Laboratory of Pharmaceutical Biotechnology and Bioinformatics, Department of Genetic Engineering and Biotechnology, Jashore University of Science and Technology, Jashore, Bangladesh, **2** Department of Pharmacy, Faculty of Biological Science and Technology, Jashore University of Science and Technology, Jashore, Bangladesh, **3** Department of Physics, Jashore University of Science and Technology, Jashore, Bangladesh

* mn.hasan@just.edu.bd

## Abstract

This study reports the first green synthesis of silver nanoparticles (AgNPs) using an aqueous leaf extract of *Brownlowia tersa* (*B. tersa*), confirming their formation via UV-Vis spectrophotometry with a peak at 472 nm. The AgNPs were evaluated for their antimicrobial and neuroprotective activities, particularly for Alzheimer's disease. GC-MS and FT-IR analyses identified phenolics and flavonoids as the capping and reducing agents in the synthesis process. FESEM imaging showed nanoparticles of varied shapes and sizes, while EDX analysis confirmed the presence of silver, oxygen, and carbon. The antimicrobial activity was demonstrated with inhibition zones ranging from 8.5 to 10.5 mm against *Staphylococcus aureus*, *Bacillus cereus*, *Shigella flexneri*, and *Pseudomonas aeruginosa* at concentrations of 100 and 200 µg/disc, compared to control, deionised water. In antioxidant assays, AgNPs exhibited an $IC_{50}$ of 1533.448 µg/ml, significantly higher than L-ascorbic acid ($IC_{50}$: 83.948 µg/ml) and the plant extract ($IC_{50}$: 254.438 µg/ml), indicating weaker antioxidant properties of AgNPs. The cytotoxicity of the AgNPs was assessed using the brine shrimp lethality test, yielding $LC_{50}$ value of 13.50 µg/ml, indicating moderate toxicity. Neuropharmacological tests, except the elevated plus maze, on swiss albino mice revealed significant anxiolytic effects ($p < 0.05$) and enhanced locomotor activity. Molecular docking studies of bioactive compounds from *B. tersa* leaves with Alzheimer's and bacterial infection-related proteins revealed binding energies from −7.7 to −5.2 kcal/mol, with Benzamide, N-ethyl-N-[(4-ethylaminophenyl) sulfonyl]- and 3,4-Dihydroxyphenylglycol showing the strongest affinities. While the results are promising, the study acknowledges challenges in scaling up the synthesis process and

**Data availability statement:** All relevant data are within the paper and its Supporting Information files.

**Funding:** The author(s) received no specific funding for this work.

**Competing interests:** The authors have declared that no competing interests exist.

emphasises the need for further research into the long-term biological effects and safety of AgNPs. Overall, *B. tersa*-derived AgNPs show great potential for therapeutic applications in bacterial infections and neurological disorders, but scalability and safety require more investigation.

## 1. Introduction

The scientific community has been very interested in nanomaterials because of their exceptional properties and diverse applications, particularly in the field of environmental remediation and biomedicine. Nanomaterials hold great potential in numerous areas, including photocatalysis, bio-imaging, photovoltaics, bio-labelling, biomedicine, solar cell technology, drug delivery, and data storage. Silver nanoparticles (AgNPs) have attracted particular interest among these because of growing worries about antibiotic resistance, the need for remedies against extremely robust bacteria, and Alzheimer's disease [1,2]. Various methods have been established for AgNP synthesis, encompassing chemical, physical, and biological approaches. However, a lot of the chemicals employed in conventional synthesis processes are expensive, hazardous, and result in by-products that are not good for the environment. Biological synthesis techniques have recently become a more attractive substitute for AgNP production [3]. Compared to conventional physical and chemical processes, the green synthesis approach has several advantages since it is less expensive, environmentally friendly, and scalable for large-scale manufacturing without the need for hazardous chemicals, pressure, high energy, or temperature [4].

Currently, there is increasing interest in adopting green synthesis methods for AgNPs, which use eco-friendly and non-toxic substances. Since green synthesis of AgNPs using algae, plants, and microbes is an environmentally friendly, biocompatible, and natural method, it is very alluring. AgNP synthesis using plant-based materials is preferable to bacterial compared to chemical processes because it is easier to implement, requires less energy, and eliminates the possibility of bacterial and dangerous chemical contamination [5]. Recent research has explored the biosynthesis of AgNPs with an emphasis on their antibacterial qualities, employing microbes and plant extracts as reducing agents. Plant extracts contain biochemicals like carboxylic acids, aldehydes, flavonoids, tannins, phenolics, ketones, and proteins that aid in the reduction of $Ag^+$ to $Ag^0$, which results in the creation of AgNP [6]. The features of biosynthesised AgNPs, including size, shape, and morphology, vary depending on the experimental conditions such as pH, adsorption of capping agents, temperature, and metal salt interaction kinetics. Thus, AgNP properties are highly dependent on these variables [3]. Consequently, the key focus in nanoparticle synthesis research is to create a method that can regulate the stability, shape, size, and physicochemical properties of AgNPs [2].

Although it is unknown how precisely plant extracts are used to synthesize nanoparticles, it is thought that plant biomolecules, including proteins, phenols, and flavonoids, are essential for stabilizing the biosynthesized nanoparticles and lowering metal ions. Researchers have shown that bio-reduction occurs in three stages:

[1] reduction of silver ions and nucleation, [2] growth and aggregation, and [3] final capping and stabilization [7]. In this process, plant phytochemicals are essential, especially secondary metabolites like proteins, polyphenols, terpenoids, phenolic acids, sugars, amides, and ketones. In many cases, the plant's reducing agents also serve as both stabilizing and capping agents [2].

AgNPs exhibit potent antibacterial properties, making them effective in preventing and treating various diseases such as cancer and bacterial infection [8]. However, their exact mechanisms against multidrug-resistant bacteria remain unclear. Studies have shown that AgNPs can damage bacterial cell walls, particularly in *E. coli*, by creating "pits" that increase cell wall permeability, leading to cell death. They also demonstrate bactericidal effects on *Staphylococcus aureus* and *Salmonella typhimurium*, with effectiveness influenced by particle size. Limited research indicates antimicrobial activity against multidrug-resistant *P. aeruginosa* [9].

Additionally, nanotechnology has improved the treatment of neurological disorders like Alzheimer's disease. In animal models, nanoparticles such as silica, titanium dioxide, zinc oxide, and silver have been utilized to lower the activity of reactive oxygen species (ROS) enzymes like catalase, SOD and GSH-Px. Intravenous nanoparticle delivery techniques, especially biodegradable ones that employ poly (lactic-co-glycolic acid) or polyethylene glycol functionalized with antibodies or oligopeptides, show promise in targeting amyloid fibrils to help prevent and treat Alzheimer's disease [10].

In this context, *Brownlowia tersa* (*B. tersa*), a plant native to the Sundarbans region of Bangladesh, was selected for the synthesis of AgNPs due to its rich phytochemical composition, including flavonoids, polyphenols, and other bioactive compounds known for their reducing and stabilizing properties. These compounds are believed to play a crucial role in the reduction of silver ions to form nanoparticles, making *B. tersa* a suitable candidate for the green synthesis of AgNPs [11]. The leaves are well-known for their medicinal effects, such as antibacterial, anti-inflammatory, analgesic, antinociceptive, antidiarrheal, antioxidant, antidiabetic, and anti-allergic properties [12]. Despite its extensive medicinal potential, an in-depth literature review indicates that *B. tersa* leaves have yet to be explored for the green synthesis of AgNPs from silver ions. This lack of prior research presents a significant gap in scientific understanding and highlights the novelty of this study. By harnessing the unique phytochemical profile of *B. tersa* leaves, we aimed to develop a sustainable approach for AgNP synthesis. The biosynthesized nanoparticles were thoroughly characterized using UV-Vis spectroscopy, FESEM, FTIR, and EDX to confirm their structural and elemental properties. Subsequently, the cytotoxicity activity, antioxidant qualities, total flavonoid and phenolic contents, and antibacterial activity of the biogenically produced silver nanoparticles were then assessed. The antibacterial efficacy was tested using the disc diffusion method against four pathogenic bacterial strains: *B. cereus*, *P. aeruginosa*, *S. aureus*, and *S. flexneri*. Additionally, an *in-vivo* evaluation has been conducted to assess the neuropharmacological potential of silver nanoparticles (AgNPs). To further support these findings, an *in-silico* analysis has been performed to investigate the bioactive compounds from *B. tersa* leaves (NPs) as possible drug candidates for the treatment of bacterial infections and Alzheimer's disease.

## 2. Method and materials

### 2.1 Plant collection

Fresh *B. tersa* leaves were gathered from Bangladesh's Sundarbans region on August 1, 2024.

### 2.2 Plant identification

The Bangladesh National Herbarium verified the specimens and provided a scientific identification. As a record of ownership, a specimen of this compilation has been given the accession number DACB 104605.

### 2.3 Extract preparation

First, the leaves were placed in a shaded area for 10 days and dried at room temperature. After the plant's leaves had air-dried, they were minced into small pieces and processed into a coarse powder using a mechanical chopper. In order to

prevent fungal growth, this powder was maintained in appropriate containers in a dark, cool atmosphere. 300 mL of 100% ethanol was used to soak about 100g of the powder for ten days, occasionally shaking and stirring. Next, Whatman Grade 1 filter paper was used to filter the amalgam. At 50 °C and 40 rpm, the crude extract was extracted using a rotary evaporator (RE-100 PRO, DLAB Scientific Inc., Beijing, China); the end yield was 2.7 g (2.7% w/w).

## 2.4 Experimental animal

Pharmacy Research Laboratory and Animal House at Jashore University of Science and Technology, Jashore-7408, Bangladesh, provided the young Swiss albino mice, male or female, that were 6−7 weeks old and weighed an average of 25−30 g. These mice were maintained under standard living conditions, which included a temperature of $27.0 \pm 1.0$ºC, a dark-light cycle of 12:12, and a relative humidity of 55−65%. They were fed rodent chow feed and allowed unlimited access to water. The mice were given a week to get used to the research setting. The work was preceded by approval of the *in vivo* experiments by the Faculty of Biological Science and Technology's Ethical Review Committee, Jashore University of Science and Technology, which ensured that all methods followed their rules [ERC/FBS/JUST/2022–110].

## 2.5 Anaesthesia, analgesia, and euthanasia procedure

**2.5.1 Anaesthesia during behavioural testing.** No anaesthesia was administered during the behavioural tests to avoid interfering with the animals' natural exploratory and locomotor behaviour. Mice were fully awake and alert throughout the testing. All measures were taken to minimise stress and discomfort during the procedure.

**2.5.2 Analgesia.** Since no surgical procedures were performed, no analgesics were required during the study.

**2.5.3 Euthanasia.** At the conclusion of the experiment, euthanasia was performed using an Isoflurane overdose. Mice were euthanized by cervical dislocation under deep anaesthesia to ensure a rapid and human death, in accordance with ethical guidelines [13].

## 2.6 Preparation of plant aqueous extract

Fresh leaves were gathered, cleaned with distilled water to remove any dust or residue, and then rinsed with deionised water to prepare the aqueous extract. 20 g of the cleaned *B. tersa* leaves were then finely chopped in 100 mL of deionised water and processed in a blender. After that, the mixture was put in a 500 mL beaker and heated to 80ºC for 60 minutes. After allowing the aqueous leaf extract to settle to ambient temperature, or about 25ºC, it was filtered through Whatman filter sheets 41 and 42, respectively. After being filtered, the leaf extracts were gathered and kept for subsequent use at 4°C [14].

## 2.7 Synthesis of AgNPs

Freshly prepared aqueous plant extract was mixed with 90 mL of a 5mM silver nitrate (90%, v/v) solution ($AgNO_3$, Fisher Scientific, Hampton, NH, USA), then it was left in direct sunshine for 10 minutes at 32°C with a solar intensity of about 72,000 lux. The color shift of the reaction mixture was used to track the synthesis of AgNPs at regular intervals of 1, 2, 3, and 10 minutes. To collect the synthesized nanoparticles, the reaction mixture was centrifuged for 15 minutes at 4ºC at 12,000 rpm. The pellets were then cleaned three times using deionized water and then dried in a vacuum drier (Yamato Scientific Co. Ltd., In Tokyo, Japan) [15].

## 2.8 Characterization of AgNPs

**2.8.1 UV-Vis spectrophotometry.** The formation of silver nanoparticles (AgNPs) was monitored using a UV-Vis spectrophotometer (Model: Perkin Elmer Lambda 950, UK). A 1 cm path length quartz cuvette was used for all measurements. After mixing 90 mL of a 5mM silver nitrate ($AgNO_3$) solution with 10 mL of the aqueous *Brownlowia*

*tersa* leaf extract, the reaction mixture was exposed to sunlight for 10 minutes. The UV-Vis spectrum was recorded at a wavelength range of 300–1100 nm to monitor the surface plasmon resonance (SPR) of the synthesized AgNPs [15].

**2.8.2 Fourier transform infrared (FTIR) spectroscopy.** FTIR analysis was conducted to identify the functional groups involved in the reduction and stabilization of AgNPs. Dried AgNPs were mixed with potassium bromide (KBr) and pressed into pellets. FTIR spectra were recorded using a Perkin Elmer Spectrum 100 FTIR Spectrometer (USA) in the range of 450–4000 cm$^{-1}$ with a resolution of 4 cm$^{-1}$. The functional groups present in the sample were analyzed by comparing the absorption peaks with reference spectra for various biomolecules [15].

**2.8.3 FESEM and EDX.** The morphology and size distribution of the AgNPs were analyzed using a Field Emission Scanning Electron Microscope (FESEM, Model: Zeiss Sigma 300, Germany). A small amount of the synthesized AgNPs was deposited onto a carbon-coated copper grid and dried at room temperature. The images were captured at various magnifications to observe the particle shape, size, and distribution. Elemental analysis was performed using Energy Dispersive X-ray (EDX) spectroscopy, integrated with the FESEM system, to confirm the presence of silver (Ag) and other elements in the sample [16].

## 2.9 Phytochemical analysis of AgNPs

**2.9.1 Total phenolic assay.** Polyphenols in plant extracts and silver nanoparticles react with Folin-Ciocalteu reagent, producing blue-colored chromogens that are quantifiable via spectrophotometric analysis. For this study, the total phenolic content (TPC) of plant extracts and silver nanoparticles was assessed using a protocol adapted from established methodologies with minor adjustments [17]. A gallic acid calibration standard was prepared across a concentration gradient (20–100 µg/mL), while the test extract was dissolved at 500 µg/mL. To initiate the reaction, 5 mL of 10% Folin-Ciocalteu reagent (Scharlab S.L., Spain) and 4 mL of 7% sodium carbonate solution (Emplura, Merck, India) were sequentially added to the extract and silver nanoparticles. After 20 minutes of incubation under ambient conditions, absorbance was recorded at 750 nm using a visible-light spectrophotometer. A linear regression curve correlating gallic acid concentration with absorbance was constructed for quantification.

The total phenolic content was calculated using the formula:

TPC, A=(C*V)/M (mg/g)

In the equation, **A** represents the phenolic measurement, *C* corresponds to the gallic acid equivalent concentration (derived from the standard calibration curve), **V** is the volume of the extract solution used, and **M** is the weight of the extract sample.

**2.9.2 Total flavonoid assay.** The total flavonoid content was determined using the aluminium chloride colorimetric method [18]. Specifically, 1 mL of the extract and silver nanoparticles (500 µg/mL) were mixed with 0.2 mL of a 10% (w/v) aluminium chloride solution, 0.2 mL of 1 M potassium acetate, and 5.6 mL of distilled water. After thorough mixing, the absorbance was measured at 415 nm against a blank. A standard calibration curve was constructed using quercetin concentrations ranging from 20-100 µg/mL, and the flavonoid content was subsequently expressed as milligrams of quercetin equivalents (QE) per gram of dried extract.

## 2.10 GC-MS analysis

The entire GC-MS analysis of AgNPs was conducted using a Claruso R690 gas chromatograph, a Claruso SQ 8C mass spectrometer (PerkinElmer, CA, USA), and an Elite-35.30m column (Elite-35.30 m length, 0.25 mm diameter, 0.25m film thickness). Pure Helium (99.999%) was infiltrated into a single µL sample in splitless mode at a flow rate of 1 mL per minute for 40 minutes. The material was characterized using high-intensity electron ionization (70eV). The column oven was first heated to 600°C for one minute, then it was raised to 2400°C by 50 °C per minute, and it stayed there for four minutes while the input temperature remained at 2800 °C. By comparing the sample's components with the National Institute of Standards and Technology (NIST) database, they can be identified [19].

## 2.11 Anti-oxidant activity

A modified method based on Reetika et al. was employed to assess the free radical scavenging activity of 1,1-Diphenyl-2-picrylhydrazyl (DPPH) using biosynthesized silver nanoparticles (aqueous) and ethanol extracts of the plant. Different concentrations of silver nanoparticles (15.625, 31.25, 62.5, 125, 250, and 500 μL/mL), plant extract (ethanol), and ascorbic acid (used as a positive control) were prepared in separate test tubes. Each sample was then given around 6mL of DPPH solution (0.004% (w/v), dissolved in ethanol), and thoroughly vortexed [20]. The resultant solutions were incubated in the dark at room temperature for 30 minutes, after that 517 nm absorbance was detected using a UV-Vis spectrophotometer (Perkin-Elmer Lambda 950, UK). A control was prepared by using DPPH without any sample, following the same procedure as described. The free radicle scavenging activity was calculated using the formula:

$$\% \text{ DPPH Scavenging Activity} = (1 - OD_{Control} / OD_{sample}) * 100 \tag{1}$$

Here $OD_{control}$ represents the absorbance of DPPH + ethanol and DPPH + deionised water for Plant extract and silver nanoparticles respectively, and $OD_{sample}$ represents the absorbance of DPPH + sample. The $IC_{50}$ value was calculated using the regression line equation which is based on the various concentrations of tested substance.

## 2.12 Cytotoxicity

The brine shrimp lethality assay was used to evaluate the cytotoxic effect of AgNPs using the Bibi et al. method [21]. Brine shrimp (*Artemia salina*) eggs were incubated in seawater (3.8% sea salt solution) until they hatched. In this experiment, different concentrations of the sample solution (15.625, 31.25, 62.5, 125, 250, and 500 μg/ml) were prepared. Twenty nauplii were meticulously counted using a sterile pipette and then put into vials with 2 mL of seawater. After adding 1 mL of the sample, more seawater was added to bring the total amount down to 10 mL. For a whole day, the vials were stored at room temperature. Double-distilled water served as the negative control, and vincristine sulfate was the positive control. To determine the $LC_{50}$ values, the best fit between concentration and lethality was displayed for each trial, which was carried out in triplicate.

$$\% \text{Morality} = \left( \frac{\text{Number of Dead Nauppii}}{\text{Number of Total Naupii Introduced}} \right) * 100 \tag{2}$$

## 2.13 *In Vitro* antibacterial activity

**2.13.1 Zone of inhibition.** Antibacterial activity against four multidrug-resistant bacteria—*S. aureus*, *B. cereus*, *P. aeruginosa*, and *S. flexneri*—was assessed using the disc diffusion method. A microbial suspension was prepared from single bacterial colony, adjusted to a turbidity of 0.5 McFarland standard. Bacteria were cultured individually on nutrient agar plates, and four discs were placed on the surface of each plate. These discs contained different concentrations of *B. tersa* leaves AgNPs (100 μg/disc, and 200 μg/disc), Negative controls (water) and antibiotics such as azithromycin (30 μg/disc) (positive controls). A volume of 10 μL from each sample was carefully applied to the disks. The plates were then incubated at 37 °C for 24 hours [22]. Following the incubation period, zones of inhibition were seen, signifying those specific areas were free of bacterial growth.

**2.13.2 Minimum inhibitory concentration (MIC).** The CLSI (2012) guidelines were followed to determine the MIC of green synthesized AgNPs [23]. MIC testing was performed using the broth microdilution technique in a 96-well round-bottom microtiter plate. A concentration of $10^6$ CFU/mL was used to standardize the bacterial inoculum. Across columns 1–8 of the microtiter plates, 100 μL of AgNPs stock solution (20 mg/mL) was serially diluted in 100 μL of nutritional broth for the MIC assay. The concentration of AgNPs was highest in column 1 and lowest in column 8. With medium,

bacterial inocula, and antibiotics (Azithromycin-3 mg/ml), column 11 served as the positive control. Column 12 serves as the negative control. 96 well plate kept in incubation for 18 hours at 37°C. After these periods, each well received an additional 30 µL of resazurin solution (0.02%) and kept for a further 3–4 hours in incubation. Afterward, Bacterial growth was indicated by a pink or colorless shift, whereas no bacterial growth was shown by a blue or purple tint. The lowest dose of AgNPs that inhibited bacterial growth while preserving the blue hue was known as the minimum inhibitory concentration (MIC) [24].

**2.13.3 Minimum bactericidal concentration (MBC).** A conventional procedure outlined by Dash et al. was used to determine the particles' minimum bactericidal concentration (MBC) [25]. MIC dilutions were sub cultured onto autoclaved nutrient agar plates; they were then incubated at 37 °C for 24 hours. The MBC was defined as the lowest concentration of nanoparticles that completely eradicated the tested bacteria. In order to determine this, the concentration at which 100% bacterial growth was suppressed was compared to an untreated positive control. In a biosafety cabinet, every experiment was carried out in a sterile setting [25].

### 2.14 *In vivo* studies

**2.14.1 Dosing groups.** For the *in vivo* experiment, 80 mice were randomly assigned into 16 groups, with 5 mice in each group. Two groups received AgNPs (20 mg/kg and 10 mg/kg, respectively), while the control group was given distilled water, and the positive control was treated with diazepam at 1 mg/kg body weight.

**2.14.2 Open field test (OFT).** In this test, locomotion, exploration, and anxiety were all measured at the same time using a slightly modified Uddin et al.,2018 approach [26]. The floor was separated into 8×8 squares of equal size, and the 45×45 cm open field apparatus alternated between white and black squares. Each mouse was immediately positioned in the centre of the apparatus after receiving an oral dose. Over 3 minutes, the total number of squares visited by each mouse was recorded at the time intervals of 0, 30, 60, 90, and 120 minutes. Each experiment was followed by wiping the open field equipment with dry tissue paper, and a 70% alcohol solution and then letting it air.

**2.14.3 Hole board test (HBT).** The hole-board test (HBT) was conducted with slight modifications following the methodology of File & Wardill et al. [27]. This is a well-established and reliable pharmacological model for assessing anxiolytic and/or anxiogenic activity. The apparatus consisted of a 40×40 cm² wooden board perforated with 16 evenly spaced holes (3 cm in diameter, 2.2 cm in depth). Mice were grouped and treated in the same manner as in the open field test. Following drug administration, each mouse was placed at the center of the board and allowed to explore freely for three minutes. The number of head dips into the holes was recorded at 0, 30, 60, 90, and 120 minutes, and these values were used to calculate the exploratory score for each mouse. The percentage of movement inhibition caused by the *B. tersa* leaves AgNPs was determined using the formula:

$$\% \text{ of movement inhibition} = (Mc - Mt)/\ Mc \times 100 \tag{3}$$

The average number of head dips in the test group is denoted by Mt, while the average number in the control group is represented by Mc.

**2.14.4 Hole cross test (HCT).** HCT was used to evaluate exploratory behavior and locomotion, as explained by Sarkar et al., 2021 [28]. A 30×20×15 cm wooden cage was used as a perforated cross. A 3 cm diameter hole was drilled in the cage's hub, and a wooden divider was fastened to the center of the cage. The mouse's spontaneous movement across the aperture from one compartment to the other (the number of crossings) was recorded for three minutes at 0, 30, 60, 90, and 120 minutes following oral administration of the test material.

**2.14.5 Elevated plus maze test (EPM).** The Elevated Plus-Maze test is an adaptation of the validated assay developed by Lister for mice [29] to assess exploratory, anxiety, and locomotor activities. Briefly, the plus-shaped EPM apparatus, made of wood, was composed of two pairs of opposite arms: two enclosed arms, 50×10×30 cm, and two open arms opposite to each other, 50×10 cm each, extending from a common central platform. It was installed at a

height of 70 cm from the ground surface level. After being handled, each mouse was identified as the moment the animal positioned all four paws on the arms. Each mouse was positioned in the middle of the maze, facing one of the enclosing arms, after receiving the control, diazepam, or treatment for 30 minutes. During a 5-minute session, the number of entries into the open arms and the duration spent there were recorded. A soundproof setting was used for the treatments. The plus maze was properly cleaned with 70% ethanol in between each experiment.

### 2.15 *In silico* studies

**2.15.1 Ligand library preparation.** 44 compounds identified from *B. tersa* leaves AgNPs via GC-MS analysis were retrieved from the PubChem database. After being imported for molecular docking experiments, all ligands were subjected to energy minimization using PyRx software.

**2.15.2 Protein preparation.** The RCSB Protein Data Bank provided the crystal structures of the following enzymes:(3R)-hydroxymyristoyl-[acyl carrier protein] dehydratase (PDB ID: 1U1Z), tyrosyl-tRNA synthetase (PDB ID: 1JIJ), amyloid A4 protein (PDB ID: 1AAP), and M4 metalloprotease protein (PDB ID: 1NPC) [30]. The protein structures were prepared for molecular docking using BIOVIA Discovery Studio. Prior to docking, polar hydrogen atoms were added to the crystal structures, while water molecules and heteroatoms were removed to minimize unwanted interactions.

**2.15.3 Molecular docking.** Molecular docking study was carried out using the AutoDock Vina module of the PyRx virtual screening tool in order to determine the binding affinities between the target protein and phytocompounds. After docking, binding energies for the protein-ligand interactions were computed and saved as CSV files for further analysis. Docked complexes were also saved in PDBQT format. The interactions of the protein-ligand complexes were visualized and analyzed in 3D and 2D structure formats using the BIOVIA Discovery Studio Visualizer and ligplot+2.2, respectively [31].

**2.15.4 Pharmacokinetics study of phytocompound.** *In-silico* prediction of Lipinski physiochemical properties and pharmacokinetic highlights of absorption, distribution, metabolism, excretion, and toxicity (ADMET) were carried out to ascertain the safety and resemblance to candidate drugs of the phytocompounds extracted from the *B. tersa* leaves AgNPs. This was done to determine whether the chemical components could be used as a novel asset of therapeutic medicines without having any negative effects on skin affectability or hepatoxicity [32]. For this objective, Swiss ADME and pKcsm online servers were utilized suitably to attain the aiming comes about.

**2.15.5 Pass activity.** The online server Prediction of Activity Spectra for Substance (PASS) (http://www.pharmaexpert.ru/passonline) was used to examine each identified molecule for antibacterial and anti-Alzheimer's activity. Based on the database, the online server forecasts the activity of small molecules. The results are shown as the likelihood that the compound will exhibit activity (Pa) or inactivity (Pi) at various pharmacological activities. Experimentally, If Pa is more than 0.7, the small molecule should be extremely active; if Pa is between 0.7 and 0.5, it should be moderately active; and if Pa is less than 0.5, it should have no biological effect [33].

### 2.16 Statistical analysis

Each experiment was conducted independently at least three times, and the results were expressed as mean±Standard deviation (SD). Software R (version 4.4.2) was utilized to execute the Welch's T-Test for *in vitro* data. Using SPSS software version 29.0 (SPSS, Chicago, IL, USA) and one-way ANOVA were used to analyze *in vivo* data. Then, Dunnett's multiple comparison test and Tukey's post hoc test were applied. P-values were considered statistically significant if they were less than 0.05.

## 3. Result

### 3.1 Green synthesis and characterization of AgNPs

**3.1.1 UV-Vis spectrometry.** The reaction mixture consisting of *B. tersa* aqueous leaves extract and $AgNO_3$ solution changed from an orange-brown color to a dark reddish-brown color (**Fig 1**) within 10 minutes. The synthesis of *B. tersa* leaves AgNPs was proved by the UV-Vis, which exhibited a distinct peak at a wavelength of 472 nm within 10 minutes (**Fig 2**).

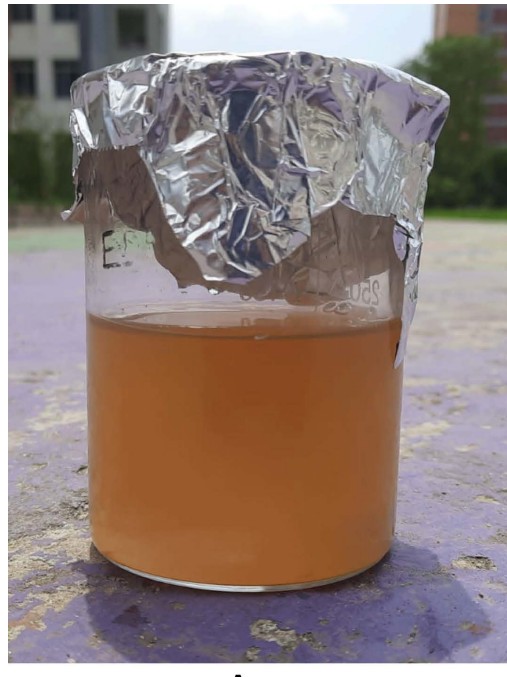
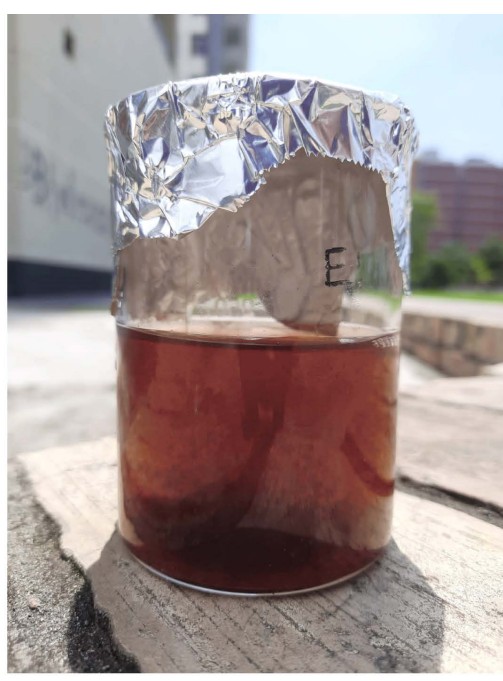

| A | B |

**Fig 1. Colour changed in the mixture, before (A) and subsequent (B) Sunshine Exposure.**

**3.1.2 FTIR Analysis.** Silver nanoparticles (AgNPs) synthesized using *B. tersa* leaves have drawn interest for their potential applications in the biomedical field. The FTIR spectra of the biosynthesized AgNPs is displayed in **Fig 3**. The analysis identified prominent peaks at 3286.50, 1590.66, 1351.70, and 1044.91 cm$^{-1}$. The strong bands at 3286.5 cm$^{-1}$ suggest the presence of alcohols with free hydroxyl (OH) groups. Peaks at 1590.66 cm$^{-1}$ correspond to the N-H functional of primary amines in proteins, while the absorption bands observed at 1351.70 and 1044.91 cm$^{-1}$ are indicative of alcohols with free hydroxyl (OH) groups, and C-N stretching of amine.

**3.1.3 FESEM/EDX analysis.** The surface structure and elemental makeup of the samples were analyzed through FESEM/EDX techniques. **Fig 4A** illustrates the morphology of silver nanoparticles synthesized from *B. tersa* leaves AgNPs, showing them as aggregates with varying shapes and sizes less than 100 nm. The elemental composition and relative abundance of the biosynthesized *B. tersa* leaves AgNPs, determined through Energy Dispersive X-ray (EDX) analysis, are displayed in **Fig 4B**. The EDX spectrum (**Fig 4B**) confirms the purity and provides a complete profile of the chemical elements present in the AgNPs. Notably, a significant proportion of silver (Ag) was detected among other elements. The EDX spectrum, which exhibited an optical absorption peak at 3 keV, indicated the relative elemental composition as follows: Oxygen (O) at 48.31%, and Silver (Ag) at 14.17%.

### 3.2 *In vitro* biological activities

**3.2.1 Phytochemical analysis.** Phytocompounds are essential in the biosynthesis of nanoparticles obtained through medicinal plants. When comparing Total Phenolic (TPC) and Total Flavonoid Content (TFC), we observed that *B. tersa* pure extract exhibited higher levels than *B. tersa* leaves AgNPs (**Table 1**). This discrepancy may be due to the phytochemicals' role in reducing silver ions to form nanoparticles, where they also function as stabilizing agents. These

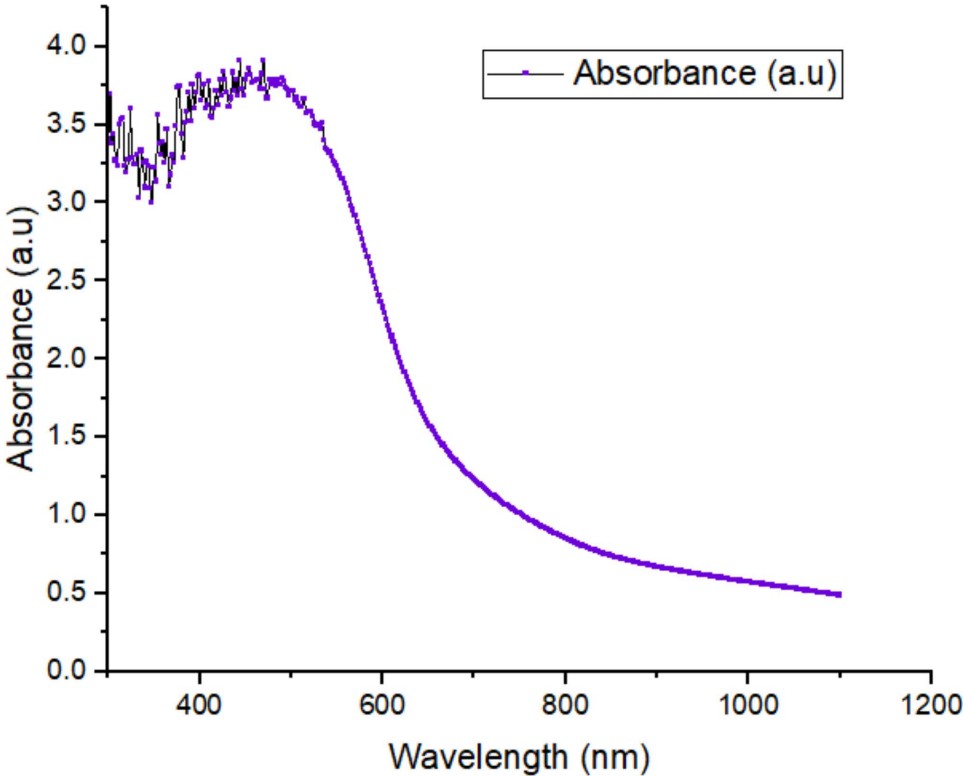

**Fig 2. UV-Vis spectra of *B. tersa* leaves-derived AgNPs, showing a characteristic metal peak at 472 nm, confirming the successful synthesis of silver nanoparticles.**

findings were further corroborated by the FTIR spectra analysis of the samples (**Fig 3**). Total phenolic content was determined from the calibration curves of gallic acid (y = 0.0083x − 0.0404, R² = 0.9987) and Total flavonoid content was determined from the calibration curves of quercetin (y = 0.004x − 0.0052, R² = 0.991) (**S1** and **S2 Figs**).

**3.2.2 DPPH assay.** The DPPH assay was conducted to evaluate the antioxidant potential of *B. tersa* leaf-derived silver nanoparticles (AgNPs). This assay measures the ability of the nanoparticles to reduce DPPH radicals to their corresponding hydrazine form by pairing the unpaired electrons, indicating antioxidant activity. The $IC_{50}$ values obtained for L-ascorbic acid, *B. tersa* leaf extract, and *B. tersa* leaf-derived AgNPs were 83.948, 254.438, and 1533.448 µg/ml, respectively, as shown in **Fig. 5**. The significantly higher $IC_{50}$ value for AgNPs indicates weaker antioxidant activity compared to L-ascorbic acid and the plant extract. This reduced antioxidant capacity of AgNPs could be attributed to the potential loss of bioactive phenolic compounds during the nanoparticle synthesis process. Not all phenolics from the leaf extract are incorporated into the nanoparticles, which may result in a decrease in antioxidant effectiveness. The DPPH radical scavenging activity was evaluated across varying concentrations of the AgNPs and the plant extract, with the results showing a clear difference in antioxidant activity, as depicted in **Fig. 5**. These findings suggest that while AgNPs possess some antioxidant activity, their potential may be limited when compared to the pure extract and the standard antioxidant, L-ascorbic acid.

**3.2.3 *In Vitro* cytotoxicity assay.** The cytotoxic effects of *B. tersa* leaves AgNPs were evaluated using a brine shrimp lethality assay. Results showed that as AgNPs concentration increased, the mortality rate of brine shrimp nauplii also rose (**Fig 6**). By plotting the mortality rate against the concentration, the $LC_{50}$ value was derived using the regression line. The

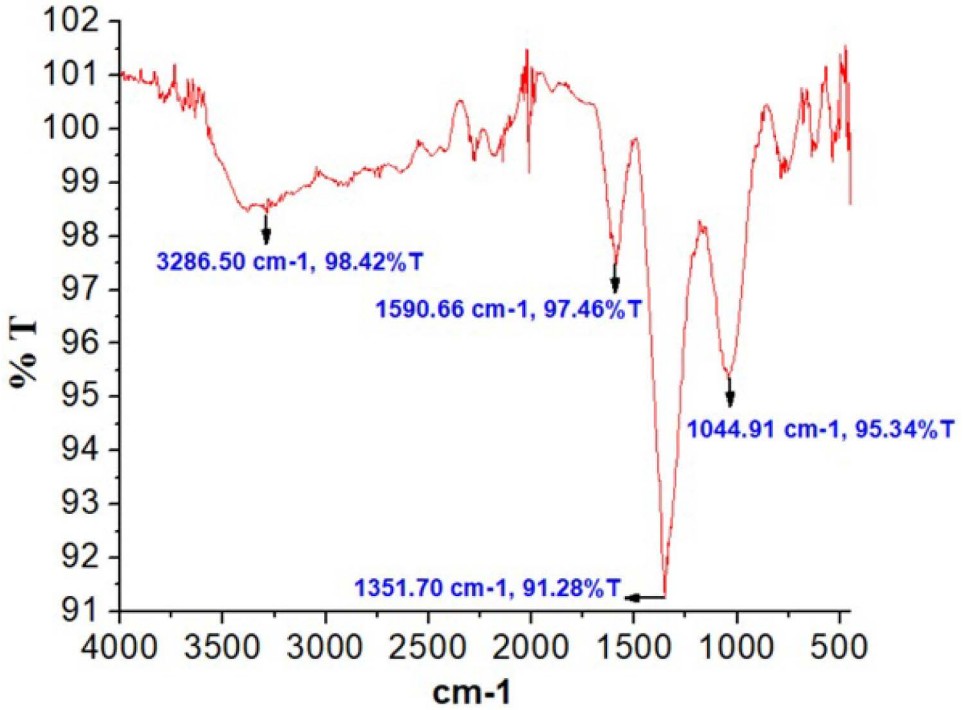

**Fig 3. FT-IR spectra of *B. tersa* leaves-derived AgNPs, showing characteristic metal peaks and functional groups (C-N, O-H, N-H), which confirm their role in the reduction and stabilization of silver nanoparticles.**

*B. tersa* leaves AgNPs demonstrated an $LC_{50}$ value of 13.49568 µg/mL, whereas vincristine sulfate $LC_{50}$ was 10.13 µg/mL. These findings indicate that AgNPs possess moderate cytotoxic activity.

**3.2.4 GC-MS analysis.** The investigation of plant-derived organic compounds and their biological activities has gained significant attention in recent years. Gas Chromatography-Mass Spectrometry (GC-MS) has become a widely used technique, combining an exceptional separation method (GC) with a highly efficient identification technique (MS), making it a preferred choice for the qualitative analysis of bioactive compounds such as semi-volatile and volatile compounds.

The chromatogram obtained through GC-MS analysis of *B. tersa* leaves AgNPs is shown in **Fig 7**. The chromatogram reveals peaks representing different chemical constituents in the sample. The y-axis shows the abundance or intensity of the compounds in relation to retention time (RT), in minutes, indicated by the x-axis, where each peak corresponds to a specific compound detected by the GC-MS system. GC-MS analysis of the *B. tersa* leaves AgNPs revealed forty-four distinct chemical compounds. Detailed information about the bioactive compounds, including their molecular weight, peak areas, and retention times, is provided in **Table 2**. Among the identified compounds, the most abundant were 3,4-DIHYDROXYMANDELIC ACID- (12.37%), HEXANEDIOIC ACID, BIS(2-ETHYLHEXYL) ESTER (12.27%), and OCTADECANE, 2,6,10,14-TETRAMETHYL- (6.93%).

### 3.3 *In vitro* Antibacterial Activity

**3.3.1 Zone of Inhibition.** The antibacterial effect of *B. tersa* leaves AgNPs against four pathogenic bacteria (*P. aeruginosa*, *S.flexneri* (gram-negative), and *B.cereus*, *S.aureus* (gram-positive) were also checked at two concentrations (100 and 200 µg/disc) (**Fig 8**). The disc diffusion method was applied, with distilled water as a negative control. The antimicrobial activity of the NPs was evaluated by measuring inhibition zone diameter. As shown in (**Fig 8A**-**8E**), AgNPs

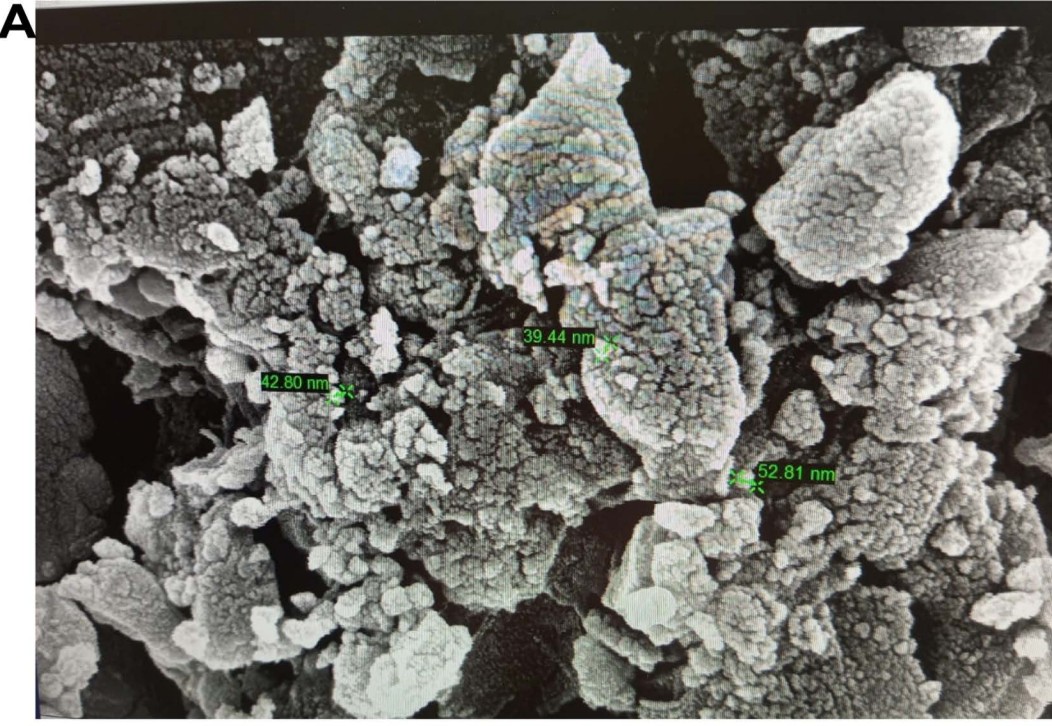

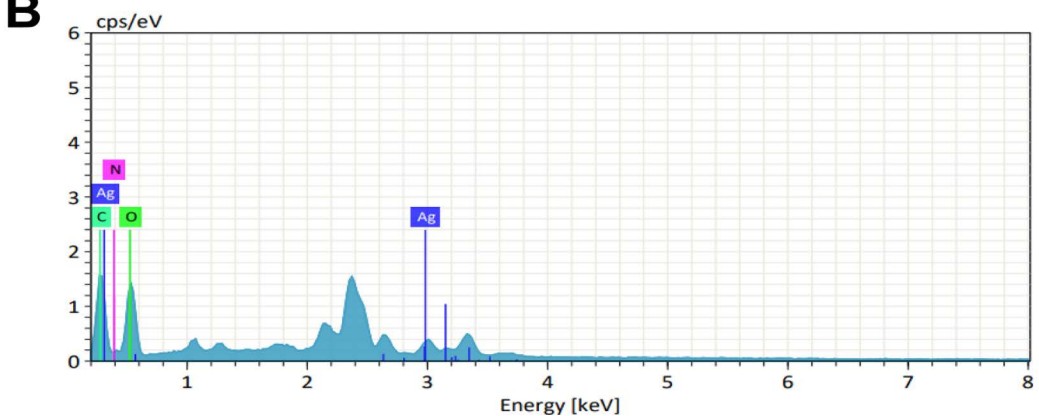

**Fig 4. A) FESEM micrograph B) EDX Spectrum.**

**Table 1. Total flavonoid and phenolic content. Data is shown as Mean±SD (n=3).**

|  | Phenolic (mg gallic acid equivalent/g sample) | Flavonoid (mg quercetin equivalent/g sample) |
|---|---|---|
| *B. tersa* | 422.63±2.90 | 49.1±1.322 |
| *B. tersa* leaves AgNPs | 48.29±2.73 | 32.43±2.08 |

exhibited notable antibacterial effects on four bacterial strains after being incubated for 24 hours at 37 °C. The inhibition zone rose as the concentration of AgNPs increased. Notably, each bacteria exhibited a larger zone of inhibition, which was consistent with the nanoparticle effects. The inhibition zone sizes and mean values for each bacterial strain, along

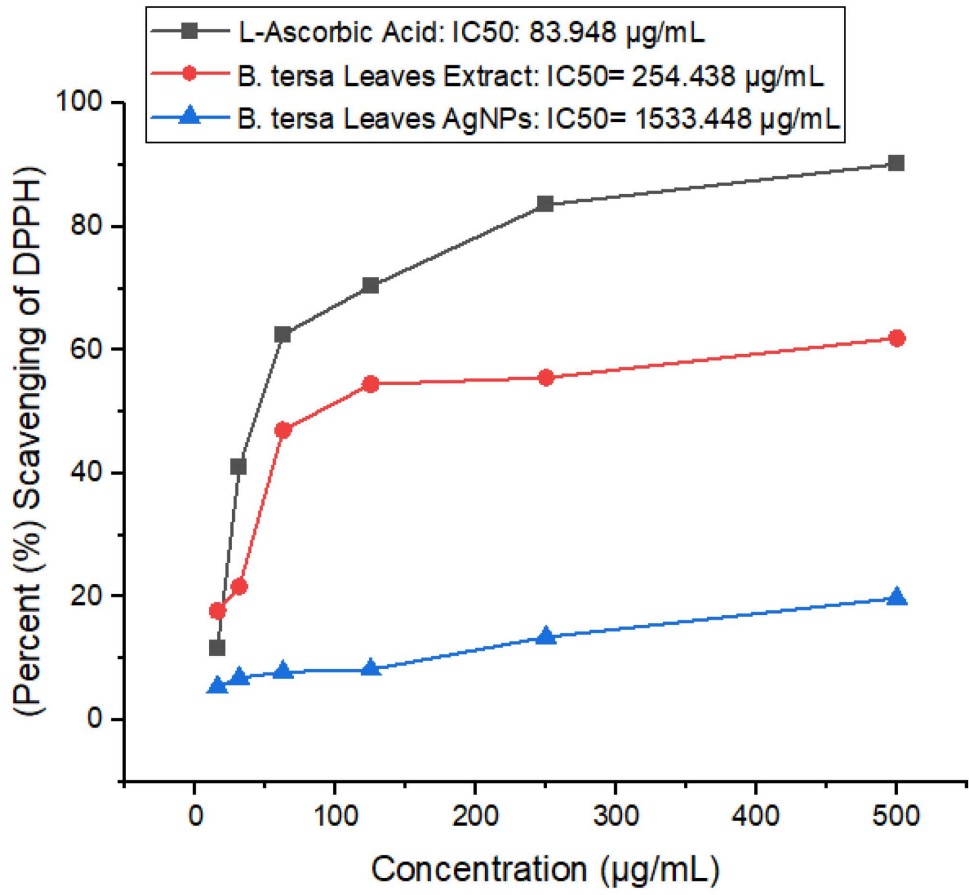

**Fig 5. Percentage inhibitions of *B. tersa* leaves silver nanoparticles and *B. tersa* leaves extract over standard ascorbic acid by DPPH assay.** Results are presented as Mean±SD (n=3), with L-ascorbic acid serving as the standard reference.

with statistical analysis, are presented in **Table 3**. The study also compared the antibacterial efficacy of AgNPs with that of a standard antibiotic Azithromycin. For *S.aureus*, AgNPs showed a dose-dependent increase in inhibition zone size, measuring 8.5±0.71 mm at 100 µg/disc and 10.5±2.12 mm at 200 µg/disc, while the standard antibiotic produced a 20±0.00 mm zone. Similarly, for *P. aeruginosa*, AgNPs exhibited inhibition zones of 8 mm at 100 µg/disc and 9 mm at 200 µg/disc, while the standard antibiotic had a 19.5±0.71 mm zone. But, for *S. flexneri*, AgNPs exhibit similar zones of inhibition of 10 mm, and 10.5 mm at 100 µg/disc and 200 µg/disc, respectively whereas *B. cereus* showed 10 mm zone of inhibition at both concentrations. Overall, this demonstrates the notable antimicrobial effects of AgNPs against the tested bacteria, focusing on their individual impact without considering any synergistic effects. The inhibition zones (in mm) for four bacterial strains, using varying concentrations of AgNPs compared to the standard antibiotic, are shown. In this study, Azithromycin was used as the standard antibiotic, with a disc concentration of 30 µg/disc.

**3.3.2 MIC and MBC.** Resazurin microtiter assay was employed to investigate the minimum bactericidal concentration and minimum inhibitory concentration of silver nanoparticles. In this method a resazurin dye is reduced to its resorufin form by mitochondrial NADH enzymes, resulting in a color change from purple to pink [34]. The study involved *P. aeruginosa*, *S.flexneri* (gram-negative), and *S.aureus*, *B.cereus* (gram-positive). Against *S. flexneri*, and *P. aeruginosa*, the MIC values of AgNPs were found to be 1.25, and 5 mg/mL, respectively (**Table 4**). For *B. cereus* and *S. aureus*, the MIC values were 1.25 mg/mL for both. The MBC values for AgNPs were determined to be 10, and 2.5 mg/mL against *P.*

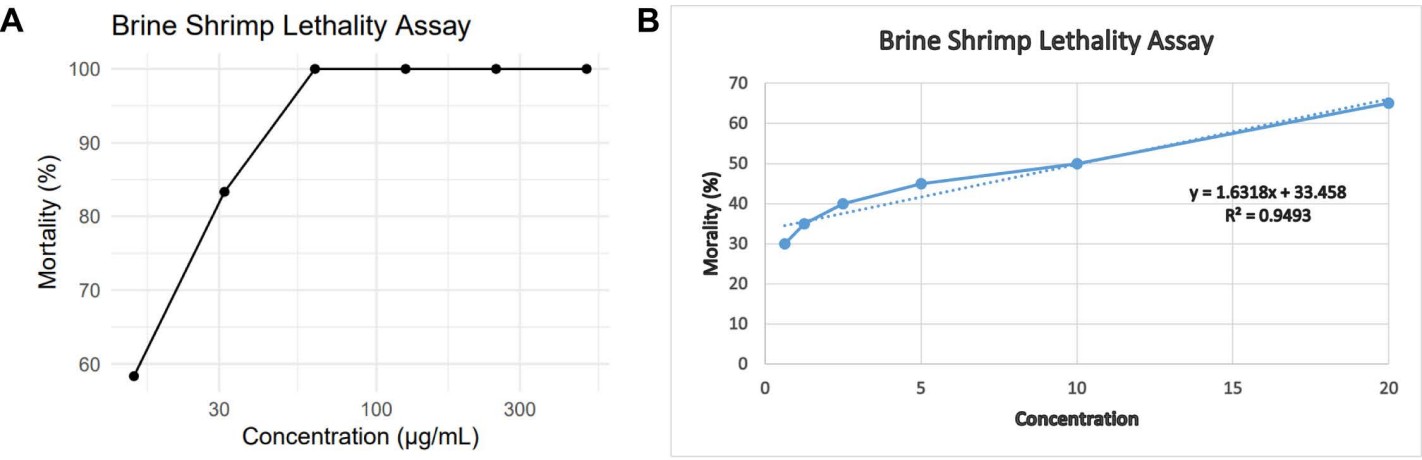

**Fig 6. The cytotoxicity of *B. tersa* leaves AgNPs. A)**. *B. tersa* leaves AgNPs **B)**. Vincristine sulfate displays the mortality rate of brine shrimp nauplii at various concentrations. All Value is expressed as mean±SD (n=3).

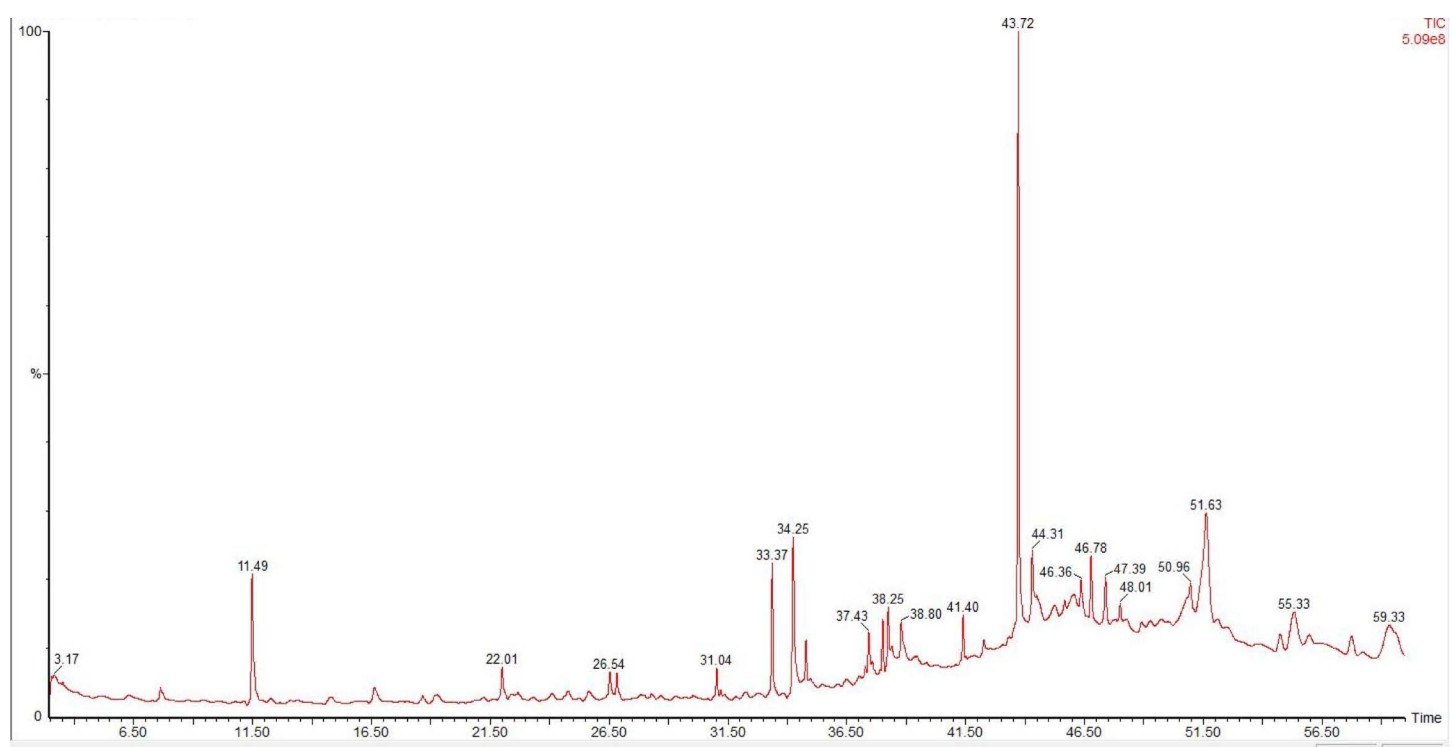

**Fig 7. GC-MS graph of *B. tersa* leaves AgNPs.**

*aeruginosa*, and *S.flexneri* respectively, and 2.5 mg/mL against *B.cereus*, and *S.aureus*. The positive control, Azithromycin (3 mg/mL) was used in this study. Pseudomonas showed slightly higher resistance than the others. Photographic evidence of the 96-well plate result is shown in **Fig 9**.

**Table 2. Analyzing *B. tersa* leaves AgNPs extract components with GC-MS.**

| S.N. | Retention Time (RT) | Name of the Compounds | Molecular Weight | %Area |
|---|---|---|---|---|
| 1 | 3.17 | 3-CHLOROPROPIONIC ACID, HEPTYL ESTER | 206 | 0.11 |
| 2 | 3.54 | PHENOL, 3-METHOXY-2-METHYL- | 138 | 0.03 |
| 3 | 7.63 | BENZYL ALCOHOL | 108 | 0.91 |
| 4 | 11.49 | BENZYLCARBAMATE | 151 | 3.926 |
| 5 | 12.28 | 3-N-HEXYLTHIANE, S,S-DIOXIDE | 218 | 0.41 |
| 6 | 14.81 | D-Mannotetradecance-1,2,3,4,5-Pentaol | 278 | 0.56 |
| 7 | 18.66 | Chloroacetic Acid, Tetradecyl Ester | 290 | 0.33 |
| 8 | 22.01 | Tetradecamethyl- | 519.07 | 0.98 |
| 9 | 22.67 | NONANOIC ACID, 9-OXO-, METHYL ESTER | 186 | 0.37 |
| 10 | 24.10 | MERCAPTOETHANOL- | 78.14 | 0.37 |
| 11 | 24.78 | DIETHYL PHTHALATE | 222 | 0.614 |
| 12 | 25.66 | BENZAMIDE, N-ETHYL-N-[(4-ETHYLAMINOPHENYL)SULFONYL]- | 332 | 0.56 |
| 13 | 26.54 | 2-AZIDOMETHYL-1,3,3-TRIMETHYL-CYCLOHEXENE | 179 | 0.62 |
| 14 | 26.83 | 3,4-DIHYDROXYPHENYLGLYCOL | 170.16 | 0.44 |
| 15 | 28.29 | METHYL 8-METHYL-NONANOATE | 186 | 0.22 |
| 16 | 28.68 | 3-ETHOXY-1,1,1,7,7,7-HEXAMETHYL- | 563.2 | 0.18 |
| 17 | 31.04 | OCTADECAMETHYL- | 607.3 | 0.66 |
| 18 | 31.21 | Z,Z-6,28-Heptatriactontadien-2-one | 530 | 0.21 |
| 19 | 32.27 | NEOPHYTADIENE | 278 | 0.51 |
| 20 | 33.37 | Tetradecanoic Acid,10,13-Dimethyl-,Methyl Ester | 270 | 3.04 |
| 21 | 34.25 | N-HEXADECANOIC ACID | 256 | 3.96 |
| 22 | 34.79 | TETRACOSAMETHYL- | 889.8 | 0.82 |
| 23 | 36.49 | 1,1,3,3,5,5,7,7,9,9,11,11,13,13,15,15-HEXADECAMETHYL- | 579.2 | 0.411 |
| 24 | 37.29 | Trans, cis-1,8-dimethylspiro [4,5] decane | 166 | 0.34 |
| 25 | 37.43 | 13-Octadecenoic Acid, Methyl Ester | 296 | 1.05 |
| 26 | 38.01 | HEPTADECANOIC ACID, 16-METHYL-, METHYL ESTER | 298 | 0.86 |
| 27 | 38.25 | HEXADECAMETHYL- | 533.1 | 1.25 |
| 28 | 38.80 | OCTADECANOIC ACID | 284 | 1.39 |
| 29 | 41.40 | 1,1,1,3,5,7,7,7-OCTAMETHYL- | 510.9 | 0.94 |
| 30 | 42.28 | HEPTACOSANOIC ACID, 25-METHYL-, METHYL ESTER | 438 | 0.87 |
| 31 | 43.72 | Hexane dioic Acid, Bis(2-ethylhexyl) Ester | 370 | 12.27 |
| 32 | 44.31 | 1,1,1,3,5,5,7,7,7-NONAMETHYL- | 322.69 | 3.16 |
| 33 | 45.67 | HENTRIACONTANE | 436 | 0.47 |
| 34 | 46.36 | OCTADECANOIC ACID, 11-METHYL-, METHYL ESTER | 312 | 1.15 |
| 35 | 46.78 | BIS(2-ETHYLHEXYL) PHTHALATE | 390 | 1.25 |
| 36 | 47.39 | EICOSAMETHYL- | 681.5 | 1.39 |
| 37 | 48.01 | EICOSANE, 2,6,10,14,18-PENTAMETHYL- | 352 | 0.77 |
| 38 | 48.91 | TRIACONTANOIC ACID, METHYL ESTER | 466 | 0.58 |
| 39 | 50.97 | OCTADECANE, 2,6,10,14-TETRAMETHYL- | 310 | 6.93 |
| 40 | 51.63 | 3,4-DIHYDROXYMANDELIC ACID- | 184.15 | 12.37 |
| 41 | 54.74 | TRIACONTANE | 422 | 0.74 |
| 42 | 55.33 | BENZALDEHYDE, 4-(DIMETHYLAMINO)-2-HYDROXY- | 165 | 3.37 |
| 43 | 55.97 | CYCLOPENTANECARBOXYLIC ACID, DODEC-9-YNYL ESTER | 278 | 2.68 |
| 44 | 57.74 | DODECAMETHYL- | 444.92 | 0.97 |

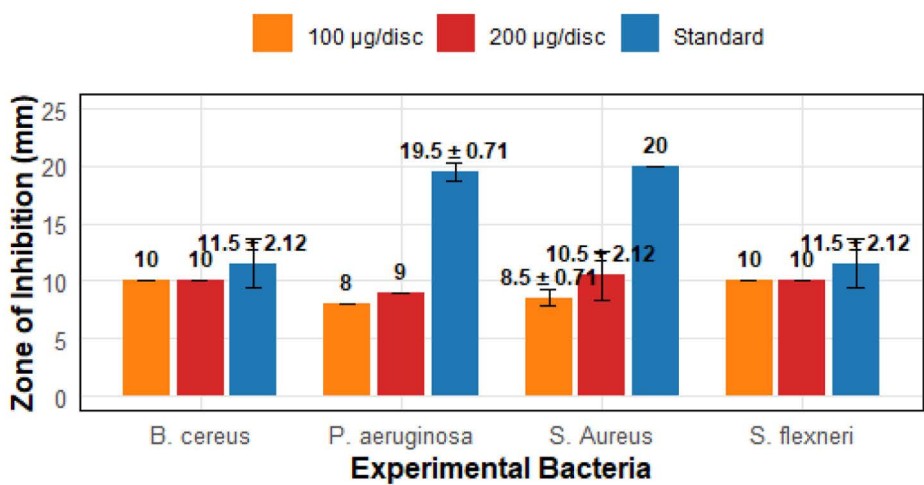

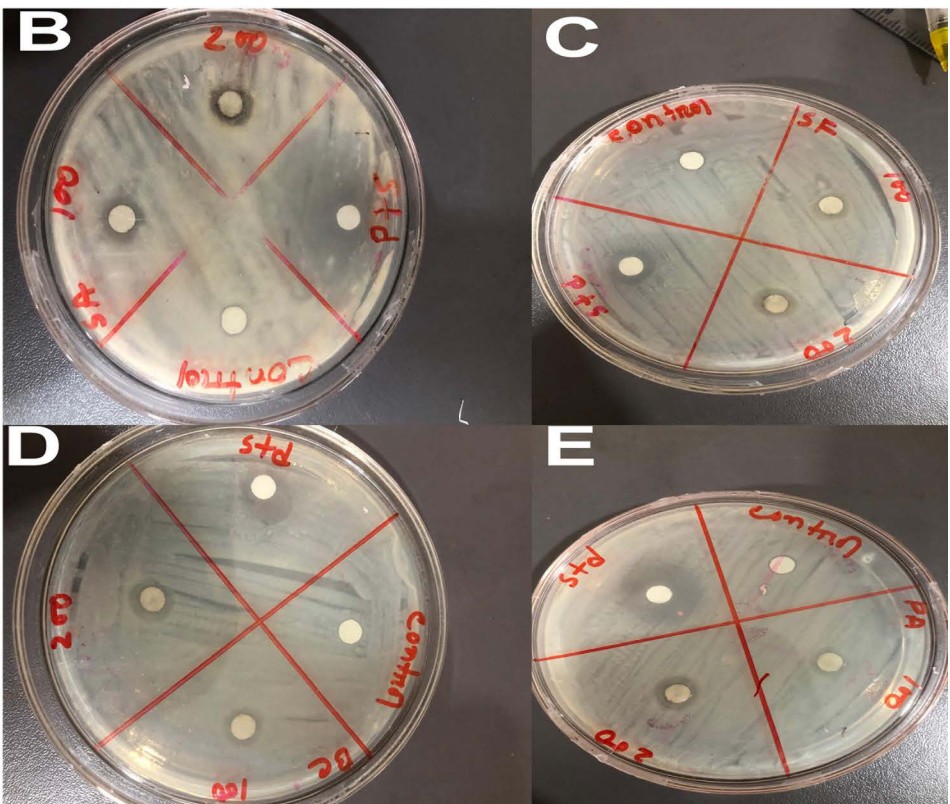

**Fig 8. Depicts the diameter of the inhibition zones for AgNPs. (B)** *S. aureus* **(C)** *S. flexneri* **(D)** *B. cereus* **(E)** *P. aeruginosa*.

**Table 3. Antibacterial activity of AgNPs. All results are stated as mean±SD (n=2).**

| Bacteria | Standard (mm) | 100 µg/disc (mm) | 200 µg/disc (mm) |
|---|---|---|---|
| *S. Aureus* | 20±0.00 | 8.5±0.71 | 10.5±2.12 |
| *S. flexneri* | 11.5±2.12 | 10.0±0.00 | 10.5±0.00 |
| *B. cereus* | 11.5±2.12 | 10.0±0.00 | 10.0±0.00 |
| *P. aeruginosa* | 19.5±0.71 | 8.0±0.00 | 9.0±0.00 |

**Table 4. MIC and MBC showed by *B. tersa* leaves AgNPs against test organisms.**
**All data are represented as mean±SD (n=3).**

| Bacteria | MIC | MBC |
|---|---|---|
| *S.aureus* | 1.25 mg/ml | 2.5 mg/ml |
| *S.flexneri* | 1.25 mg/ml | 2.5 mg/ml |
| *B. cereus* | 1.25 mg/ml | 2.5 mg/ml |
| *P.aeruginosa* | 5 mg/ml | 10 mg/ml |

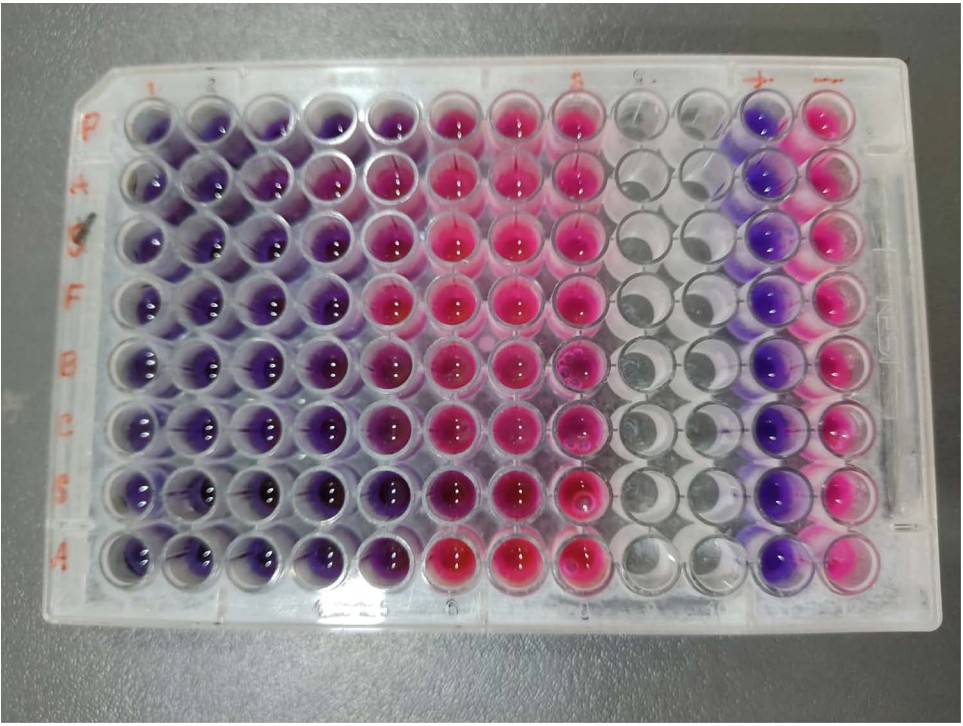

**Fig 9. MIC of Ag NPs against (First two row)** *P. aeruginosa* **(Second two row)** *S. flexneri* **(Third two row (***B.cereus***) and (Last two row)** *S. aureus* **shown in 96 well plates.**

### 3.4 In vivo

**3.4.1 Open field test.** The open-field test is employed to assess sedative and exploratory behaviors. *B. tersa* leaves AgNPs administered at a dose of 10 mg/kg, notably decreased locomotor activity during both the second (30 minutes) and third (60 minutes) observation periods in comparison to the control group (p<0.05). Similarly, a dose (20 mg/kg)

demonstrated a progressive decline in locomotor activity starting from the second observation period, which persisted through the fourth period, demonstrating a significant difference when compared to the control (p<0.05) (**Fig 10**).

### 3.4.2 Hole cross test.

The hole cross test was employed to assess the CNS depressant effects of *B. tersa* leaves AgNPs. At a dose of 10 mg/kg, AgNPs exhibited a gradual decline in movement during the second (30 minutes) to fifth (120 minutes) observation intervals compared to the control group. Similarly, the 20 mg/kg dose displayed variable effects on movement throughout the observations, with the highest level of movement recorded during the first observation (**Fig 11**).

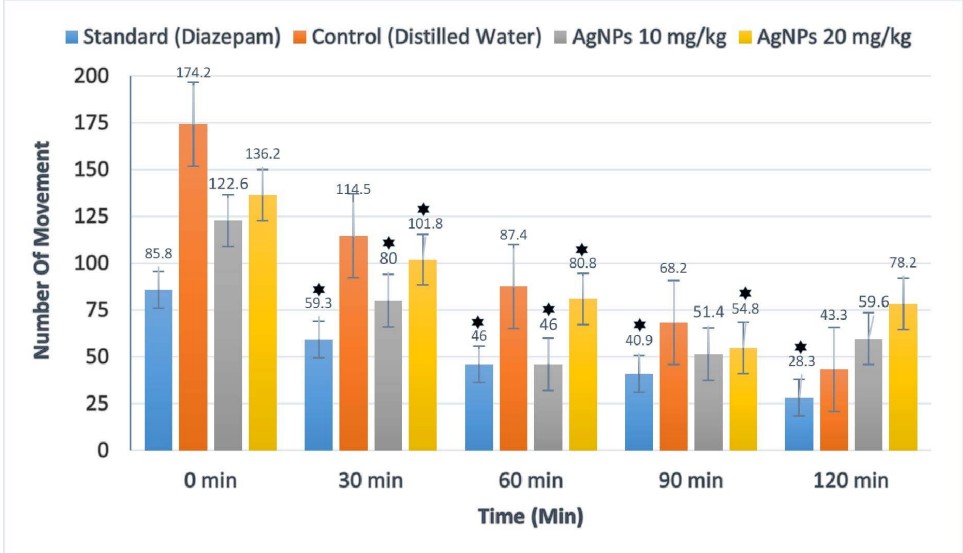

**Fig 10. Effects of *B. tersa* leaves AgNPs in the open field test.** Value is expressed as mean±SD (n=5), with * p<0.05.

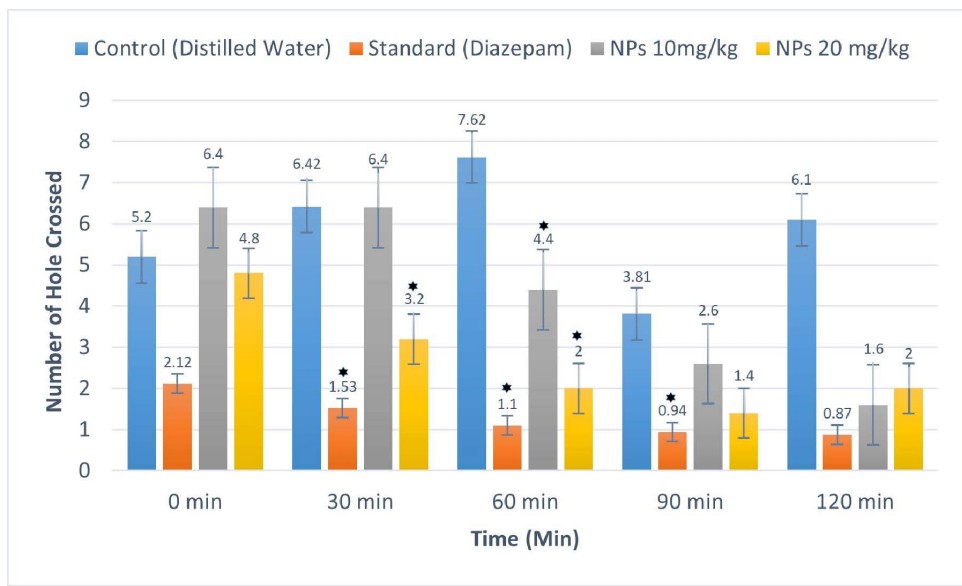

**Fig 11. Effects of *B. tersa* leaves AgNPs in hole cross test.** Data is displayed as mean±SD, where n=5. *P<0.05, vs. control (distilled water, 10 mL/kg BW).

**3.4.3 Hole board test.** In the hole-board test, mice administered with 10 mg/kg and 20 mg/kg of *B. tersa* leaves AgNPs exhibited changes in their head-dipping behavior over time. As shown in **Fig 12**, 10 mg/kg induce significantly more head dips at 30 min compared to the control group (*$P < 0.05$). However, the 20 mg/kg dose led to a marked rise (*$P < 0.05$) in head dips at both 30 and 60 minutes, with values of 21.8 ± 12.70 and 24.2 ± 12.21, respectively, compared to the control group.

**3.4.4 Elevated Plus Test.** In the Elevated Plus Maze test, the control group spent 30.0 ± 2.0 seconds in the open arm, while the AgNPs 10 mg/kg and AgNPs 20 mg/kg groups spent 32.4 ± 5.8 and 33.6 ± 7.9 seconds, respectively. The Diazepam group, however, showed a significant increase in open arm time (236.1 ± 10.0 seconds, $p < 0.001$) (Fig 13A). For closed arm time, the control group spent 270.0 ± 3.0 seconds, and both AgNPs 10 mg/kg (267.8 ± 4.2) and AgNPs 20 mg/kg (271.5 ± 3.8) groups did not show significant differences compared to the Control. In contrast, Diazepam significantly reduced the time spent in the closed arm to 60.8 ± 2.5 seconds ($p < 0.001$) (Fig 13B). Although a trend toward increased open arm time was observed with AgNPs, these differences were not statistically significant, suggesting that AgNPs at both doses did not significantly alter anxiety-related behavior compared to Control, while Diazepam exhibited a clear anxiolytic effect.

### 3.5 *In silico* study

**3.5.1 Molecular docking analysis.** The study revealed the molecular interactions and binding affinities of identified phytocompounds with four target receptors: Amyloid A4 protein (PDB ID: 1AAP), tyrosyl-tRNA synthetase (PDB ID: 1JIJ), (3R)-hydroxymyristoyl-[acyl carrier protein] dehydratase (PDB ID: 1U1Z), and M4 metalloprotease protein (PDB ID: 1NPC). The ligands were successfully docked into the predicted binding sites of the receptors, where the grid box was centered at x = 25.90, y = 18.64, z = 44.29, with grid dimensions of 60 × 60 × 60, and with the binding strength evaluated based on the scoring function and lowest binding energy. A summary of the findings is provided in **Table 5**. The binding energy values ranged from −6.4 to −5.2 kcal/mol, −7.1 to −5.9 kcal/mol, −7.7 to −5.9 kcal/mol, and −6.6 to −4.8 kcal/mol for 1AAP, 1JIJ, 1U1Z, and 1NPC, respectively. Among the results, the compound Benzamide, N-ethyl-N-[(4-ethylaminophenyl)

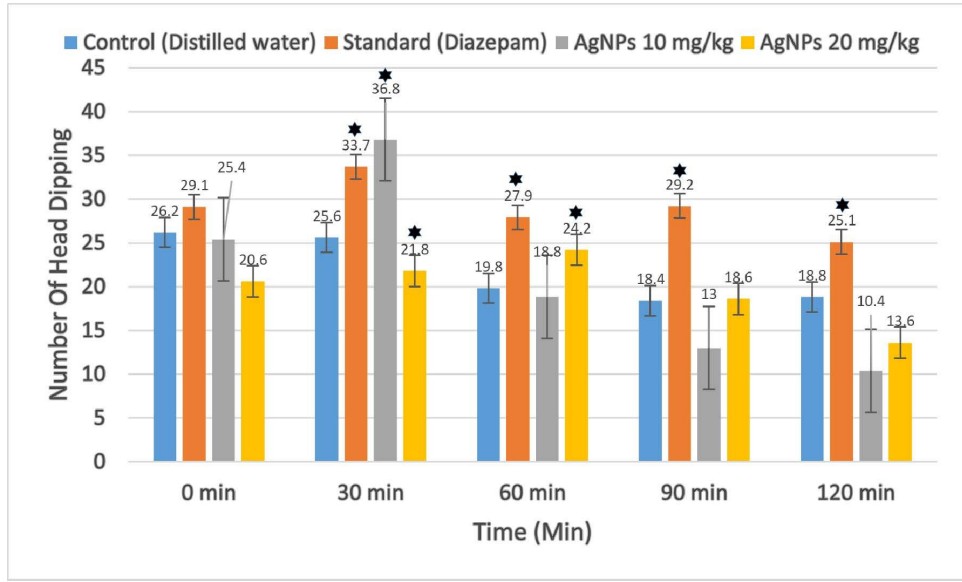

**Fig 12. Neuropharmacological Activity of *B. tersa* leaves AgNPs in hole board test.** Values are represented as mean ± SD, where n = 5. *$P < 0.05$.

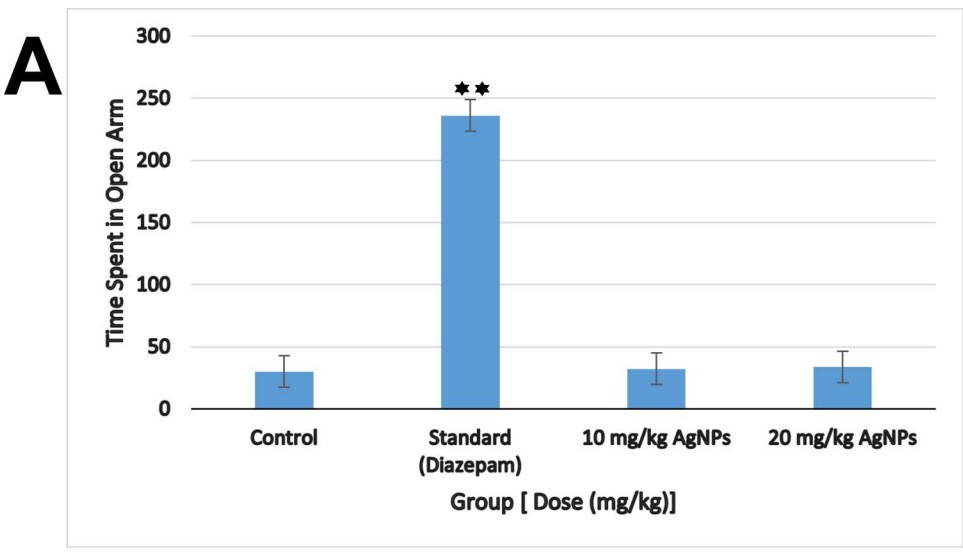

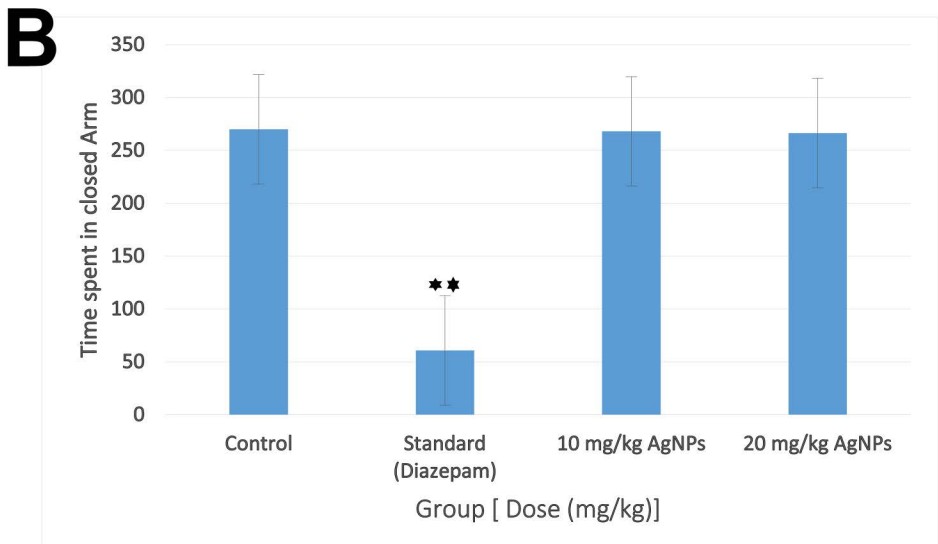

**Fig 13. Time spent in A) open arm B) Closed arm, Effects of *B. tersa* leaves AgNPs (10 mg/kg and 20 mg/kg) in elevated plus maze test.** Data is displayed as mean±SD, where n=5. *P<0.05.

sulfonyl]- demonstrated the highest binding affinity, with binding energies of −6.4 kcal/mol for 1AAP and the compounds 3,4-Dihydroxyphenylglycol showed −7.0 kcal/mol for 1JIJ. **Table 5** shows the docking results of ligands including standard drugs (Rivastigmine for Alzheimer's disease, and azithromycin for bacterial) with the target proteins, while **S3**-**S6 Figs** show the amino acid interactions at the active sites of the receptors. The binding interactions included hydrogen bonds and hydrophobic bonds formed between the amino acid of the receptor and ligands as shown in **Table 5**. Based on the molecular docking analysis, it is proposed that phytochemical compounds from *B. tersa* leaves AgNPs extract could serve as potential analogs for developing anti-Alzheimer and anti-bacterial agents.

**3.5.2 ADMET analysis.** The ADMET methodology was employed to identify promising new medicinal compounds by evaluating their effectiveness and safety. ADMET, which stands for absorption, distribution, metabolism, excretion, and toxicity,

**Table 5. The intermolecular interaction and the binding affinity between macromolecule Amyloid A4 protein (PDB ID: 1AAP), tyrosyl-tRNA synthetase (PDB ID: 1JIJ), (3R)-hydroxymyristoyl-[acyl carrier protein] dehydratase (PDB ID: 1U1Z), and M4 metalloprotease protein (PDB ID: 1NPC) and ligands.**

| Ligands PubChem CID Number | Compounds Name | Docking Score (Kcal/mol) | Amino Acid Involved Interaction | |
|---|---|---|---|---|
| | | | **Hydrogen Bond Interaction** | **Hydrophobic Bond Interaction** |
| **Ligands for Amyloid A4 protein (PDB ID: 1AAP)** | | | | |
| CID: 91714447 | Benzamide, N-ethyl-N-[(4-ethylaminophenyl)sulfonyl]- | −6.4 | Ala9-3.20Å, Ala9-2.88 Å, Thr26-2.86Å | Thr26, Gln8, Tyr22, Gln8 (A), Gln8 (B), Pro32, Asp24 (A), Asp24 (B), Tyr22 |
| CID:565495 | trans, trans-1,8-Dimethylspiro[4.5] decane | −5.8 | | Thr26 (A), Gln8 (A), Gln8 (B), Tyr22 (A), Ala9 (A), Ala9 (B), Thr26 (B), Asp24 (A), Asp 24 (B), Tyr22 (B) |
| CID:12136 | BENZYLCARBAMATE | −5.8 | Thr26-2.94Å, Gln8-2.90Å, Ala9-3.26Å | Gln8, Tyr22 (A), Asp24 (A), Asp24 (B), Tyr22 (B), Ala9 |
| CID: 578430 | 2-Azidomethyl-1,3,3-trimethyl-cyclohexene | −5.3 | Thr26-2.89Å, Ala9-3.24Å, Gln8-3.13Å, Asp24-2.92Å | Val25, Gln8, Tyr22 (A), Tyr22 (B), Asp24 |
| CID: 77991 (Control) | Rivastigmine | −5.2 | Tyr22-2.86Å | Thr26 (A), Pro32, Ala9 (A), Gln8 (A), Gln8 (B), Tyr22, Ala9 (B), Thr26 (B), Asp24, Phe33 |
| **Ligands for tyrosyl-tRNA synthetase (PDB ID: 1JIJ)** | | | | |
| CID: 447043 (Control) | Azithromycin | −7.1 | Val224-3.23 Å, Asp195-3.22 Å | Gly193, Gly49, Leu223, Lys234, Ala239, Trp241, His47, Pro222, His50, Pro53 |
| CID: 91528 | 3,4-Dihydroxyphenylglycol | −7.0 | Thr75-3.02 Å, Thr75-3.05 Å, Gly38-2.87 Å, Val191-3.06 Å, Asn124-2.82 Å, Asp40-3.06 Å, Tyr170-2.77 Å | Asp177, Gln196, Ile200, Gln190, Tyr36, Leu70, Gln174 |
| CID: 8343 | Bis(2-ethylhexyl) phthalate | −6.8 | Tyr170-3.18 Å | Gln174, Gly192, Val191, Glln196, Gly38, Phe54, Pro53, Gln190, Tyr36, Gly49, Gly193, Ile200, His50, Pro222, Asp195, Lys84, Asp40, Asp80, Asp177, Ala39, Leu70 |
| CID: 552347 | D-Mannotetradecane-1,2,3,4,5-pentaol | −5.9 | Asp-2.93 Å, Tyr36-3.06 Å, Gly38-3.09 Å, Gly38-3.15 Å, Asp80-2.87 Å, Gln174-3.05, Tyr170-3.07, Tyr170-2.74 Å, Gln196-3.05 Å | Leu70, Gly192, Phe54, Ala39, Gly193, Cys37, Pro53, Asp195, His50 |
| CID: 7641 | Hexanedioic acid, bis(2-ethylhexyl) ester | −5.9 | Thr166-3.24 Å | Leu133, Ile78, Thr169, Thr75, Ile131, Met77, Leu173, Gly129, Leu128, Arg125, Gly74, Phe136 |
| **Ligands for (3R)-hydroxymyristoyl-[acyl carrier protein] dehydratase (PDB ID: 1U1Z)** | | | | |
| CID: 447043 (Control) | Azithromycin | −7.7 | Gly48-3.18 Å, Phe-3.21 Å, Pro44-2.86 Å, Tyr15-2.93 Å | Tyr15 (B), Asn42, Tyr15 (A), Tyr15 (D), Tyr15 (C), Pro44 (D), Pro16 (E) |
| CID: 8343 | Bis(2-ethylhexyl) phthalate | −6.5 | Tyr15-3.20 Å, Tyr15-3.03 Å | Pro44 (B), Tyr15 (B), Pro16 (A), Asn47, Tyr15 (D), Pro16 (D), Pro44 (C), Tyr15 (C), Pro16 (B), Pro44 (D), Pro44 (E), Pro44 (A), Pro16 (C) |
| CID: 5364506 | 13-Octadecenoic acid, methyl ester | −6.1 | Tyr15-2.94 Å | Pro16 (D), Pro16 (C), Tyr15 (D), Asn47(C), Pro44 (C), Pro44 (D), Phe45 (D) |
| CID: 10446 | Neophytadiene | −6.0 | | Pro44 (A), Tyr15 (D), Tyr15 (B), Pro44 (C), Pro16 (D), Pro44 (D), Tyr15 (C), Pro16 (C), Tyr15 (A), Pro16 (A), Pro44 (B), Pro16 (B) |
| CID: 521556 | OCTADECANE, 2,6,10,14-TETRAMETHYL- | −5.9 | | Leu86 (C), His49 (D), Tyr88 (C), Ile55 (D), Gln99 (D), Val101 (D), Val39 (D), Pro100 (D), Lys37 (D), Leu60 (D), Leu107 (D), Ala64 (D), Ile61 (D), Leu19 (D), Phe45 (D), Phe46 (D), Pro57 (D), Gly58 (D), Phe89 (C), His13 (C), Ala66 (C), Tyr87 (C), Glu63(C) |

*(Continued)*

**Table 5.** (Continued)

| Ligands for Amyloid A4 protein (PDB ID: 1AAP) | | | | |
|---|---|---|---|---|
| Ligands PubChem CID Number | Compounds Name | Docking Score (Kcal/mol) | Amino Acid Involved Interaction | |
| | | | Hydrogen Bond Interaction | Hydrophobic Bond Interaction |
| Ligands for M4 metalloprotease protein (PDB ID: 1NPC) | | | | |
| CID: 447043 (Control) | Azithromycin | −6.6 | Glu144-2.92 Å, His147-3.09 Å, Trp116-3.23 Å | His143, Glu167, Asn117, His232, Tyr111, Phe115, Tyr158, Leu156, Asn166, Glu151 |
| CID: 91528 | 3,4-Dihydroxyphenylglycol | −6.0 | Asn166-3.00 Å, Asn166-3.14 Å, Asn166-2.84 Å, His147-3.09 Å, His147-3.09 Å, Trp116-3.03 Å | Tyr158, Glu144, Phe115, Glu151 |
| CID: 552347 | D-Mannotetradecane-1,2,3,4,5-pentaol | −5.4 | Glu144-2.70 Å, Glu144-3.13 Å, Asp171-2.73 Å, His143-2.92 Å, Glu167-2.84 Å, Glu167-3.07 Å, His147-3.05 Å, Tyr158-3.17 Å, His232-2.89 Å, Asp227-2.89 Å, Asp227-2.70 Å | Phe131, Val140, Leu134, Asn113, Arg204, Leu203 |
| CID: 8343 | Bis(2-ethylhexyl) phthalate | −5.4 | Tyr158-3.14 Å | Phe115, Glu151, Asn166, Trp116, His147, His232, Asn113, Tyr111 |
| CID: 7641 | Hexanedioic acid, bis(2-ethylhexyl) ester | −4.8 | | Asn113, Glu144, Tyr158, Leu156, His147, Glu151, Thr150, Asn166, Trp116, Phe115 |

is a modern approach to drug discovery and development. It focuses on the pharmacokinetic and toxicological characteristics of potential drug candidates. The designed compounds were assessed for their ADMET profiles and compliance with Lipinski's Rule of Five using tools such as the SwissADME server (http://www.swissadme.ch) and pkcsm. The findings are detailed in **Table 6**.

The Lipinski Rule of five states that a molecule might have good oral bioavailability if it fulfils four conditions: molecular mass < 500 Daltons; hydrogen bond acceptors <10; hydrogen bond donors <5; and Log P (CLogP) < 5. The compounds presented in **Table 6** align with these criteria, suggesting their potential as viable drug candidates.

Pharmacokinetic evaluations revealed that almost all compounds exhibit high intestinal absorption rates. Additionally, blood-brain barrier (BBB) permeability analysis showed that compounds CID: 91714447 (Benzamide, N-ethyl-N-[(4-ethylaminophenyl) sulfonyl]-), CID:565495 (trans, trans-1,8-Dimethylspiro [4.5]decane), CID:12136 (BENZYLCARBAMATE), CID: 578430 (2-Azidomethyl-1,3,3-trimethyl-cyclohexene), CID: 77991 (Control) (Rivastigmine) could cross the BBB, whereas other compounds could not.

Toxicity evaluations, including hepatotoxicity, AMES, and Rat Oral Acute Toxicity assessments presented in **Table 6**, confirmed that almost all compounds exhibited no toxic effects.

**3.5.3 QSAR analysis.** A total of 44 phytocompounds were analyzed using the PASS online tool (http://www.way2drug.com/passonline) to assess their potential for antibacterial and anti-Alzheimer activity. Among them, four phytocompounds for Alzheimer's disease, and seven phytocompounds for antibacterial, were selected based on their combined activities, including neurodegenerative diseases treatment, Acetylcholine neuromuscular blocking agent, Acute neurologic disorders treatment, Anti-infective and Antibacterial. Compounds with higher Pa values were considered to have greater medicinal efficacy and potential for experimental application. Through our QSAR model analysis, the top eleven phytocompounds were identified using a Pa cut-off value of ≥ 100 (**Table 7**). While PASS cannot predict binding affinity for novel therapeutic targets, it is useful in reducing the potential side effects of molecules [19]. These eleven selected phytocompounds were chosen after completing molecular docking, following an evaluation of their ADMET profiles.

## 4. Discussion

This research utilized a green synthesis approach for producing AgNPs from *B. tersa* leaves extract, highlighting the methods as both eco-friendly and economical. Silver nanoparticles are recognized for their remarkable physicochemical and biological attributes, which make them suitable for numerous therapeutic applications. This study involved the synthesis

**Table 6. Conducting ADMET profiling to analyse the pharmacological and physicochemical characteristics of the selected ligand for Alzheimer's disease and antibacterial agents.**

| Physicochemical | | | | | | | | | Pharmacological | | | | | | |
|---|---|---|---|---|---|---|---|---|---|---|---|---|---|---|---|
| ligand | MoW | HAc | HD | NRB | MoR | SA | NLV | DL | IA | BBB | TC | AT | LD50 | HT | MDT |
| CID: 91714447 Benzamide, N-ethyl-N-[(4-ethylaminophenyl)sulfonyl]- | 332.42 g/mol | 3 | 1 | 7 | 91.07 | 136.545 | 0 | Yes | High | Yes | 0.93 | Yes | 2.387 | No | 0.731 |
| CID:565495 (trans,trans-1,8-Dimethylspiro[4.5]decane | 166.30 g/mol | 0 | 0 | 0 | 55.31 | 76.742 | 1 | Yes | Low | Yes | 1.084 | No | 1.591 | No | 0.049 |
| CID:12136 (BENZYLCARBAMATE) | 151.16 g/mol | 2 | 1 | 3 | 40.60 | 64.776 | 0 | Yes | High | Yes | 0.476 | No | 2.162 | No | 1.082 |
| CID: 578430 (2-Azidomethyl-1,3,3-trimethyl-cyclohexene) | 179.26 g/mol | 2 | 0 | 3 | 52.16 | 79.462 | 0 | Yes | High | Yes | 0.074 | No | 3.117 | No | 0.338 |
| CID:77991 (Control) (Rivastigmine) | 250.34 g/mol | 3 | 0 | 6 | 73.12 | 109.146 | 0 | Yes | High | Yes | 0.557 | No | 3.402 | No | 0.382 |
| CID: 447043 (Control) (Azithromycin) | 748.98 g/mol | 14 | 5 | 7 | 200.78 | 311.558 | 2 | Yes | Low | No | −0.424 | No | 2.769 | Yes | 1.027 |
| CID: 91528 (3,4-Dihydroxyphenylglycol) | 170.16 g/mol | 4 | 4 | 2 | 42.58 | 69.338 | 0 | Yes | High | No | 0.132 | No | 1.716 | No | 1.114 |
| CID: 8343 (Bis(2-ethylhexyl) phthalate) | 390.56 g/mol | 4 | 0 | 16 | 116.30 | 170.550 | 1 | Yes | High | No | 1.898 | No | 1.451 | No | 1.393 |
| CID: 552347 (D-Mannotetradecane-1,2,3,4,5-pentaol) | 278.39 g/mol | 5 | 5 | 12 | 75.22 | 115.455 | 0 | Yes | High | No | 1.954 | No | 1.494 | No | 1.559 |
| CID: 7641 (Hexanedioic acid, bis(2-ethylhexyl) ester) | 370.57 g/mol | 4 | 0 | 19 | 110.44 | 160.953 | 1 | Yes | High | No | 1.997 | No | 1.427 | No | 0.717 |
| CID: 5364506 (13-Octadecenoic acid, methyl ester) | 296.49 g/mol | 2 | 0 | 16 | 94.26 | 131.894 | 1 | Yes | High | No | 1.978 | No | 1.637 | No | 0.04 |
| CID:10446 (Neophytadiene) | 278.52 g/mol | 0 | 0 | 13 | 97.31 | 128.294 | 1 | Yes | Low | No | 1.764 | No | 1.473 | No | 0.272 |
| CID:521556 (OCTADECANE, 2,6,10,14-TETRAMETHYL-) | 310.60 g/mol | 0 | 0 | 15 | 107.87 | 142.403 | 1 | Yes | Low | No | 1.664 | No | 1.568 | No | 0.163 |

"LD50: oral rat acute toxicity, mg/kg; NRB: No. of rotatable bonds; BBB: blood-brain barrier; MW: molecular weight; TC: total clearance, log mL/(min·kg); HD: No. of hydrogen bond donor; HAc: No. of hydrogen bond acceptor; NLV: No. of Lipinski's rule violations; HT: hepatotoxicity; AT: AMES toxicity; MTD: maximum tolerated dose for a human, log mg/(kg·day); DL: drug-likeness; IA: intestinal absorption"

of AgNPs from *B. tersa* leaves, their characterization using several biophysical techniques, and an assessment of their effects on four pathogenic microbes. This study also investigates their possible effectiveness in treating Alzheimer's disease in animal models.

To characterize the biosynthesized *B. tersa* leaves AgNPs, several techniques such as EDX, FTIR, FESEM, and UV-Vis analyses were employed. The successful formation of silver nanoparticles was confirmed by UV-Vis spectroscopy, which exhibited a distinct surface plasmon resonance (SPR) peak at 472 nm (Fig 2). This characteristic peak is a definitive indicator of the reduction of silver ions (Ag⁺) to metallic silver (Ag⁰) and reflects the collective oscillation of conduction electrons on the nanoparticle surface. Furthermore, FT-IR analysis provided critical insights into the phytochemicals responsible for the biosynthesis. The spectrum (Fig 3) showed prominent peaks at 3286.50, 1590.66, 1351.70, and 1044.91 cm$^{-1}$. The strong absorption at 3286.5 cm$^{-1}$ indicates the presence of alcohols with free hydroxyl (OH) groups.

**Table 7. The outcome of QSAR models in predicting bioactivity for ligand assessment.**

| Serial No | Compound CID | Pa | Pi | Activity |
|---|---|---|---|---|
| 1 | CID: 91714447 | 0.372 | 0.028 | Antiparkinsonian, rigidity relieving |
| | | 0.367 | 0.040 | Botulin neurotoxin A light chain inhibitor |
| | | 0.338 | 0.051 | Neuropeptide Y2 antagonist |
| 2 | CID:565495 | 0.657 | 0.010 | Neurodegenerative diseases treatment |
| | | 0.625 | 0.015 | Acetylcholine neuromuscular blocking agent |
| | | 0.584 | 0.015 | Neurotransmitter antagonist |
| 3 | CID:12136 | 0.531 | 0.065 | Acute neurologic disorders treatment |
| | | 0.479 | 0.088 | Acetylcholine neuromuscular blocking agent |
| | | 0.415 | 0.083 | Neurotransmitter antagonist |
| 4 | CID: 578430 | 0.573 | 0.047 | Acute neurological disorders treatment |
| | | 0.505 | 0.071 | Acetylcholine neuromuscular blocking agent |
| | | 0.354 | 0.129 | Neurotransmitter antagonist |
| 5 | CID:77991 (Control) | 0.500 | 0.075 | Acetylcholine neuromuscular blocking agent |
| | | 0.483 | 0.068 | Neurotransmitter uptake inhibitor |
| | | 0.346 | 0.036 | Antineurogenic pain |
| 6. | CID:447043 (Control) | 0.985 | 0.001 | Antiinfective |
| | | 0.964 | 0.000 | Antibacterial |
| | | 0.941 | 0.000 | Antibiotic |
| 7. | CID: 91528 | 0.538 | 0.019 | Antiinfective |
| | | 0.306 | 0.058 | Antibacterial |
| | | 0.288 | 0.060 | Antiparasitic |
| 8. | CID: 8343 | 0.435 | 0.035 | Antiinfective |
| | | 0.321 | 0.050 | Antiprotozoal (Coccidial) |
| | | 0.225 | 0.098 | Antibacterial |
| 9. | CID: 552347 | 0.680 | 0.008 | Antiinfective |
| | | 0.345 | 0.057 | Antimycobacterial |
| | | 0.329 | 0.050 | Antibacterial |
| 10. | CID: 7641 | 0.471 | 0.027 | Antiinfective |
| | | 0.245 | 0.124 | Antimycobacterial |
| | | 0.131 | 0.096 | Antibacterial, ophthalmic |
| 11. | CID: 5364506 | 0.541 | 0.011 | Antihelmintic (Nematodes) |
| | | 0.470 | 0.019 | Antiparasitic |
| | | 0.472 | 0.027 | Antiinfective |
| 12. | CID:10446 | 0.363 | 0.040 | Antibacterial |
| | | 0.240 | 0.165 | Antiinfective |
| | | 0.114 | 0.042 | Antineoplastic antibiotic |
| 13. | CID:521556 | 0.455 | 0.030 | Antiinfective |
| | | 0.433 | 0.024 | Antiparasitic |
| | | 0.322 | 0.053 | Antibacterial |

"Probability of activity (**Pa**) and probability of inactivity (**Pi**)"

The peak at 1590.66 cm$^{-1}$ is attributed to N–H stretching of primary amines in proteins, while the bands at 1351.70 and 1044.91 cm$^{-1}$ correspond to free hydroxyl (OH) groups of alcohols and C–N stretching vibrations of amines. The presence of these functional groups confirms that biomolecules in the *B. tersa* leaf extract, play a dual role: they act as reducing

agents to convert Ag⁺ to Ag⁰ and as capping agents to stabilize the formed nanoparticles [35]. Evidence of the synthesized nanoparticles was demonstrated by the FESEM images (**Fig 4**), which primarily showed spherical particles of diverse sizes and shapes smaller than 100 nm, though some bigger particles were also visible. The thorough analysis of aggregates revealed that there was no direct physical contact between the nanoparticles, suggesting that phytochemical components were crucial in stabilizing the AgNPs [36]. EDX analysis supported the detection of silver, alongside an indicative optical absorption peak, with additional signals from elements like C and O, probably deriving from the carbon-coated copper grid or X-ray emissions linked to proteins and enzymes found in the *B. tersa* leaves [37].

The high phenolic content of *B. tersa* leaves extract is responsible for the effective production of AgNPs. The potent reducing and antioxidant properties of phenolic compounds, make them suitable for AgNP synthesis. Because phenolic components provide electrons, the high phenolic concentration in *B. tersa* leaves extract helps reduce silver ions to nanoscale silver particles. Furthermore, quinoid chemicals produced by the oxidation of phenolic groups can stick to the surface of nanoparticles and help stabilize them in suspension. Previous research suggests that phenolic compounds possess robust antioxidant properties due to their electron-donating abilities and their role as singlet oxygen quenchers [38]. The nanoparticles formed from the leaf extract exhibit antioxidant activity, which is likely due to the phenolic compounds capping their surface. Our synthesized AgNPs exhibited superior antioxidant activity compared to the nanoparticles derived from *Phoenix sylvestris* L. seeds, as reported by Qidwai et al. [39]. The antioxidant properties of nanoparticles are reported to offer significant advantages over traditional antioxidant delivery systems, such as encapsulated protection of antioxidant agents, increased bioavailability, and the ability for targeted and controlled delivery [40]. Additionally, the synthesized *B. tersa* leaves AgNPs demonstrated significant toxicity towards brine shrimp naupii, with an $LC_{50}$ value of around 13.49568 µg/mL, compared to vincristine sulfate $LC_{50}$ of 10.13 µg/mL (**Fig 6**). This suggests that nanoparticles may contain toxic secondary metabolites, potentially linked to their anticancer properties. Similarly, Green-synthesized Ag/AgCl-NPs have been shown to have similar cytotoxic effects on brine shrimp nauplii in earlier experiments, with an $LC_{50}$ of 30 µL/mL [41].

This study assessed the biologically produced AgNPs' antibacterial efficacy against a variety of pathogenic microbes, including both positive and negative Gram strains, resulting in discernible zones of inhibition. In our results clearly indicated that the green synthesized Ag NPs shows effective potential antibacterial activity in the gram-positive bacteria compared to gram negative bacteria. The results of this study highlighted varying inhibition zones, reflecting differences in antibacterial effectiveness. The antibacterial effects of silver nanoparticles occur via different pathways, including binding to thiol groups of bacterial enzymes, intercalation with negatively charged DNA, and disrupting cell wall synthesis [42]. Silver nanoparticles accumulate on the cell surface, forming noticeable protrusions. Due to the presence of sulfur-containing proteins in the cell wall, silver nanoparticles continuously release silver ions, which exhibit a strong electrostatic attraction to sulfur, allowing them to adhere to the cell wall and plasma membrane. This interaction can lead to increased cell wall permeability, ultimately causing cell rupture. Upon entering the cytoplasm, silver nanoparticles interfere with ATP synthesis and DNA replication, leading to the inhibition of respiratory enzymes and the generation of reactive oxygen species (ROS). This oxidative stress contributes to microbial cell death. Additionally, silver ions may inhibit protein synthesis by inactivating ribosomes. During the green synthesis of silver nanoparticles, biological extracts act as capping agents, stabilizing the nanoparticles and potentially altering their dissolution properties. These capping agents may enhance the antimicrobial activity of silver nanoparticles by modifying their surface characteristics, thereby increasing their ability to disrupt microbial cells. The combined effect of silver nanoparticles and these bioactive capping molecules may further amplify their antibacterial properties [43]. Several factors influence the antibacterial efficacy of silver nanoparticles, including their size, shape, surface charge, and the type of capping agent used during synthesis. Furthermore, bacterial strains' resistance to nanoparticles is significantly influenced by their genetic characteristics; resistance can occasionally result from genetic alterations that reduce the generation of Reactive Oxygen Species (ROS) in low-oxygen environments, therefore lowering oxidative stress. Nevertheless, oxidative stress brought on by these modest ROS levels may result in mutations and increased bacterial resistance to AgNPs [42].

To substantiate antibacterial activity, minimum inhibitory concentration (MIC) and minimum bactericidal concentration (MBC) are necessary. MIC determination identifies the concentration of nanoparticles required to prevent bacterial growth. As shown in **Table 4**, AgNPs exhibited MIC values of approximately 1.25 mg/mL against Gram-positive bacteria such as *B. cereus*, and *S. aureus*. For Gram-negative bacteria like *P. aeruginosa*, and *S. flexneri*, MIC values varied for 5 and 1.25 mg/mL respectively, with *P. aeruginosa* showing lower susceptibility. This reduced susceptibility could be due to the neutralization of nanoparticle charge by the LPS layer in *P. aeruginosa*, making it less vulnerable to the nanoparticles [44]. Therefore, our AgNPs showed higher efficacy against *S. aureus,* than reported by Pimploy Ngamsurach et al. [45].

In addition, the restrictive nature of the BBB poses a challenge in treating neurological disorders by limiting drug delivery to the CNS. Metallic nanoparticles (NPs) offer a potential solution, but toxicity concerns hinder their use. Green synthesis presents a safer alternative, utilizing natural bio-reductants for enhanced biocompatibility. Green-synthesized AgNPs show promise in improving BBB penetration while reducing toxicity [46,47]. This study investigates the potential of *B. tersa* leaves AgNPs in treating anxiety and depression through behavioral assessments in a mouse model. This study employed several tests, the Hole Board Test (HBT) and Elevated Plus Maze test (EPM) were two of the tests of these to measure anxiety levels. The EPM test is commonly utilized to investigate the anxiolytic effects of drugs in mice and serves as a standard instrument for screening novel benzodiazepine-like compounds [13]. Using this approach, the behavioral response of mice treated with varying doses of AgNPs was evaluated. Surprisingly, none of the doses caused any noticeable changes in the animals' anxiety levels when tested with the Elevated Plus Maze, failing to support the expected anxiolytic effects of the AgNPs.

The open arms of the EPM are naturally anxiety-provoking for rodents, which generally leads to limited exploration of these areas. Anxiolytics promote exploration of open arms, while anxiogenics reduce it. The HBT evaluates an animal's reaction to a new environment, which can cause anxiety. According to earlier studies, this test is a valid way to gauge emotional states in these kinds of situations. Our findings demonstrated that both doses of AgNPs led to increased head dipping behavior, particularly at the 20 mg/kg dose, suggesting a reduction in anxiety. Anxiety can result from abnormal neurotransmitter activity, such as serotonin, dopamine, or GABA receptor dysfunction, or from disruptions in glutamatergic, serotonergic, GABAergic, or noradrenergic signaling [48]. In this study, we suggest that *B. tersa* leaves AgNPs may modulate chemical signaling pathways to produce their anxiolytic effects.

Locomotor activity, defined as the ability to move from one position to another, is a crucial aspect of exploratory behavior in mice and is regularly observed in animal models. Although movement is often linked to the animal's inclination to explore its environment, this behavior does not always equate to genuine exploration. Other factors, beyond the exploratory drive, can influence locomotor activity [49]. Both locomotor activity and exploratory behavior are commonly measured using behavioral tests like the Hole Board test and Open Field Test. Although motion is the main parameter of interest, other parameters like motor output, health problems, exploratory transitions, timing of the circadian rhythm, and freezing or fear-related reactions may also have a role. The total distance traveled, time spent moving, rearing activity, and behavioral changes over time are important indicators that are usually tracked. Defecation, central region time, and activity during the first five minutes are findings that might also indicate anxiety or other emotional states [50].

In the Open Field Test, throughout the 120-minute observation period, there was a steady decrease in movement, with experiment groups exhibiting significantly lower levels of locomotor activity than controls. These reductions were evident at all tested doses of *B. tersa* leaves AgNPs (10 and 20 mg/kg) and diazepam (1 mg/kg, intraperitoneally) during successive observation intervals (30, 60, for 10 mg/kg and 30, 60, and 90 minutes). Similarly, in the Hole Cross Test, *B. tersa* leaves AgNPs administration led to a gradual decrease in movement, as indicated by the reduced number of hole crossings starting from the second observation session to the fourth observation (30, 60, and 120 minutes).

According to the study's findings, *B. tersa* leaves AgNPs show promise as a source of medicinal compounds to treat Alzheimer's disease and bacterial infections. Computational approaches are crucial in the search for novel lead compounds since they not only save a great deal of time and money when compared to conventional clinical trials, but they

also drastically reduce expenses [13]. An essential technique for analysing the interactions between ligands and their target proteins is molecular docking. It sheds light on the underlying mechanisms of different pharmacological actions by revealing how tiny compounds interact with target protein active sites [32]. Our research included virtual screening of 44 bioactive compounds from the *B. tersa* leaves AgNPs extract via GC-MS analysis, targeted against AMYLOID A4 PROTEIN (PDB: 1AAP), tyrosyl-tRNA synthetase (PDB ID: 1JIJ), (3R)-hydroxymyristoyl-[acyl carrier protein] dehydratase (PDB ID: 1U1Z), and M4 metalloprotease protein (PDB ID: 1NPC) to support the conclusions drawn from *in vivo* and *in vitro* experiments. The docking analysis identified 11 compounds out of 44 with remarkable binding affinity to the four receptor proteins, as detailed in **Table 5**.

A powerful screening method called ADMET profiling evaluates factors such as molecular weight, lipophilicity, hydrogen bond acceptors and donors, and other toxicity traits. The identification of appropriate drug candidates is further improved by pharmacokinetic and pharmacophore tests; **Table 6** lists important characteristics that are essential for drug discovery. Every examined molecule exhibited drug-like properties; the majority had high intestinal absorption rates, while a few had low absorption. Conventional toxicity poses health hazards. A significant variation in $LD_{50}$ values was noted, indicating low toxicity levels for the bioactive compounds, with no hepatotoxicity and AMES toxicity except in Benzamide, N-ethyl-N-[(4-ethylaminophenyl) sulfonyl]-. Hence, compounds intended for neurological applications must effectively traverse the blood-brain barrier (BBB), which restricts solutes from entering the central nervous system. As per **Table 6**, four compounds exhibited the ability to cross the BBB and could be applied in treating Alzheimer's disease, while the remaining compounds are suitable for antibacterial use. Additionally, all compounds met the criteria for drug-likeness without breaching Lipinski's rule of five. Moreover, QSAR analysis revealed possible anti-Alzheimer and antibacterial activities. Therefore, the combined evidence from these analysis, *in vivo* neurological evaluations, *in vitro* bacterial activity studies, and *in silico* assessments supports the potential use of *B. tersa* leaves AgNPs in managing bacterial infections and mental disorders.

## 5. Conclusion

This study demonstrates the green synthesis of silver nanoparticles (AgNPs) from *Brownlowia tersa* leaf extract and evaluates their antimicrobial and neuroprotective potential. While B. tersa leaves-derived AgNPs exhibit promising biological activity *in vitro* and *in vivo*, certain challenges remain before their large-scale application. The AgNPs displayed significant biological effects, as well as moderate neuroprotective potential in various behavioral tests. However, they did not show significant anxiolytic effects in the Elevated Plus Maze test, which suggests that the neuropharmacological effects might be limited in this specific assay.

Key barriers include ensuring the consistency of nanoparticle properties, such as size and stability, across different batches during scaling up the synthesis process. Additionally, long-term monitoring is essential to assess nanoparticle stability, aggregation, and potential toxicity over extended periods of use. While the nanoparticles exhibited significant biological activity, their *in vivo* safety profile, including bioaccumulation and chronic exposure effects, requires thorough investigation.

Future research should focus on *in vivo* validation to better understand the therapeutic efficacy, bioavailability, and safety of AgNPs. Further, formulation studies are needed to optimize controlled release mechanisms and enhance the therapeutic potential of AgNPs. In addition, safety evaluations and regulatory assessments are critical to ensure AgNPs meet the necessary safety standards for clinical and commercial use, thus ensuring their viability for real-world applications. Addressing these challenges will help to fully unlock the potential of AgNPs derived from B. tersa in medical and industrial applications.

## Supporting information

**S1 Fig. Standard curve of gallic acid.**
(TIF)

**S2 Fig. Standard curve of quercetin.**
(TIF)

**S3 Fig. The depiction shows of the interaction between selected phytocompounds and Amyloid A4 protein (PDB ID: 1AAP).** On one side, we have the three-dimensional complex of protein-ligand interaction, and on the other, we have the two-dimensional complex. A. Benzamide, N-ethyl-N-[(4-ethylaminophenyl) sulfonyl]- Amyloid A4 protein (PDB ID: 1AAP) B. trans, trans-1,8-Dimethylspiro [4.5] decane – Amyloid A4 protein (PDB ID: 1AAP) C. BENZYLCARBAMAT- Amyloid A4 protein (PDB ID: 1AAP) D. 2-Azidomethyl-1,3,3-trimethyl-cyclohexene - Amyloid A4 protein (PDB ID: 1AAP) E. Rivastigmine- Amyloid A4 protein (PDB ID: 1AAP).
(TIF)

**S4 Fig. The depiction shows of the interaction between selected phytocompounds and Tyrosyl-tRNA synthetase (PDB ID: 1JIJ).** On one side, we have the three-dimensional complex of protein-ligand interaction, and on the other, we have the two-dimensional complex. A. Azithromycin- Tyrosyl-tRNA synthetase (PDB ID: 1JIJ) B. 3,4-Dihydroxyphenylglycol- Tyrosyl-tRNA synthetase (PDB ID: 1JIJ) C. Bis(2-ethylhexyl) phthalate- Tyrosyl-tRNA synthetase (PDB ID: 1JIJ) D. D-Mannotetradecane-1,2,3,4,5-pentaol - Tyrosyl-tRNA synthetase (PDB ID: 1JIJ) E. Hexanedioic acid, bis(2-ethylhexyl) ester – Tyrosyl-tRNA synthetase (PDB ID: 1JIJ).
(TIF)

**S5 Fig. The depiction shows of the interaction between selected phytocompounds and (3R)-hydroxymyristoyl-[acyl carrier protein] dehydratase (PDB ID: 1U1Z).** On one side, we have the three-dimensional complex of protein-ligand interaction, and on the other, we have the two-dimensional complex. A. Azithromycin- (3R)-hydroxymyristoyl-[acyl carrier protein] dehydratase (PDB ID: 1U1Z) B. Bis(2-ethylhexyl) phthalate- (3R)-hydroxymyristoyl-[acyl carrier protein] dehydratase (PDB ID: 1U1Z) C. 13-Octadecenoic acid, methyl ester-(3R)-hydroxymyristoyl-[acyl carrier protein] dehydratase (PDB ID: 1U1Z) D. Neophytadiene- (3R)-hydroxymyristoyl-[acyl carrier protein] dehydratase (PDB ID: 1U1Z) E. OCTADECANE, 2,6,10,14-TETRAMETHYL- (3R)-hydroxymyristoyl-[acyl carrier protein] dehydratase (PDB ID: 1U1Z).
(TIF)

**S6 Fig. The depiction shows of the interaction between selected phytocompounds and M4 metalloprotease protein (PDB ID: 1NPC).** On one side, we have the three-dimensional complex of protein-ligand interaction, and on the other, we have the two-dimensional complex. A. Azithromycin- M4 metalloprotease protein (PDB ID: 1NPC) B. 3,4-Dihydroxyphenylglycol- M4 metalloprotease protein (PDB ID: 1NPC) C. D-Mannotetradecane-1,2,3,4,5-pentaol-M4 metalloprotease protein (PDB ID: 1NPC) D. Bis(2-ethylhexyl) phthalate- M4 metalloprotease protein (PDB ID: 1NPC) E. Hexanedioic acid, bis(2-ethylhexyl) ester- M4 metalloprotease protein (PDB ID: 1NPC).
(TIF)

**S1 File. Some experimental raw data.**
(ZIP)

## Author contributions

**Conceptualization:** Md Ridoy Hossain, Md. Nazmul Hasan (Ph.D).

**Data curation:** Md Ridoy Hossain, Md Al Saber, Md. Nazmul Hasan Zilani, Md. Nazmul Hasan (Ph.D).

**Formal analysis:** Md Ridoy Hossain, Md Al Saber, Md. Anisul Hoque, Md. Shamsur Rahman, Florence Bornali Ratno, Md. Nazmul Hasan Zilani, Md Ohiduzzaman, Md. Nazmul Hasan (Ph.D).

**Investigation:** Md Ridoy Hossain, Md Al Saber, Md. Anisul Hoque, Md. Shamsur Rahman, Florence Bornali Ratno, Md. Nazmul Hasan Zilani, Md Ohiduzzaman, Md. Nazmul Hasan (Ph.D).

**Methodology:** Md Ridoy Hossain, Md Al Saber, Md. Anisul Hoque, Md. Shamsur Rahman, Florence Bornali Ratno, Md. Nazmul Hasan Zilani.

**Project administration:** Md. Nazmul Hasan Zilani, Md. Nazmul Hasan (Ph.D).

**Resources:** Md Ridoy Hossain, Md Al Saber, Md. Nazmul Hasan Zilani, Md Ohiduzzaman, Md. Nazmul Hasan (Ph.D).

**Software:** Md Ridoy Hossain, Md Al Saber, Md. Nazmul Hasan Zilani, Md Ohiduzzaman, Md. Nazmul Hasan (Ph.D).

**Supervision:** Md. Nazmul Hasan Zilani, Md. Nazmul Hasan (Ph.D).

**Validation:** Md Ridoy Hossain, Md Al Saber, Md. Nazmul Hasan Zilani, Md Ohiduzzaman, Md. Nazmul Hasan (Ph.D).

**Visualization:** Md Ridoy Hossain, Md. Nazmul Hasan Zilani, Md. Nazmul Hasan (Ph.D).

**Writing – original draft:** Md Ridoy Hossain.

**Writing – review & editing:** Md Al Saber, Md. Nazmul Hasan Zilani, Md Ohiduzzaman, Md. Nazmul Hasan (Ph.D).

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
