## [Decision Letter · Decision Letter 0]

28 May 2025

Dear Dr. Hasan (Ph.D),

Thank you for submitting your manuscript to PLOS ONE. After careful consideration, we feel that it has merit but does not fully meet PLOS ONE’s publication criteria as it currently stands. Therefore, we invite you to submit a revised version of the manuscript that addresses the points raised during the review process.

**kindly make sure the revised version is modified as per the reviewer comments and meet the journal criteria for abstract formation, number of references and language style.**

 reviewer comments are mentioned below:

We look forward to receiving your revised manuscript.

Kind regards,

Sana Shamim

Academic Editor

PLOS ONE

3. To comply with PLOS ONE submissions requirements, in your Methods section, please provide additional information regarding the experiments involving animals and ensure you have included details on (1) methods of sacrifice, (2) methods of anesthesia and/or analgesia, and (3) efforts to alleviate suffering.

5. Please remove your figures from within your manuscript file, leaving only the individual TIFF/EPS image files, uploaded separately. These will be automatically included in the reviewers’ PDF.

Additional Editor Comments:

As per the comments received from the reviewers, the article needs Major revision.

Reviewers' comments:

Reviewer's Responses to Questions

**Comments to the Author**

1. Is the manuscript technically sound, and do the data support the conclusions?

Reviewer #1: No

Reviewer #2: Yes

2. Has the statistical analysis been performed appropriately and rigorously?

Reviewer #1: No

Reviewer #2: Yes

3. Have the authors made all data underlying the findings in their manuscript fully available?

Reviewer #1: Yes

Reviewer #2: Yes

4. Is the manuscript presented in an intelligible fashion and written in standard English?

Reviewer #1: No

Reviewer #2: Yes

Reviewer #1: The authors should clarify the following sections to avoid confusion

• On page 10, line 222, authors should revise first sentence to make it sensible!

• On page 13, line 286, how the 80 mice were divided into four groups having 5 mice in each group?

• A large number of references are used in this manuscript which is not as norms of “original manuscript” writing! References should be updated with recent ones only and limit up to 50 or less!

• Table 1 show very unusual results regarding phenolic contents; need for counter check or valid discussion should be incorporated with reported evidences.

• Antioxidant activity clearly indicated negative results of synthesized nanoparticles which are not properly interpreted in results!

• Antibacterial activity is also not remarkable. Although it shows almost same zone of inhibition against 2 selected bacteria as that of standard but why not use advance available antibiotic as standard? Moreover, the synthesized nano formations not show any improved results in increasing concentrations!

• How the results of elevated plus test are significantly higher as mentioned by authors! Wrong interpretation and unsatisfactory discussion are done regarding this activity.

The authors aim to demonstrate Green Synthesis of Silver Nanoparticles from Brownlowia tersa Leaf Extract and their Multifaceted Evaluation of Antibacterial, Antioxidant, Cytotoxic, and Anti-Alzheimer Potential however, the data does not fully support the conclusion.

While the study appears to be sound, the language is unclear, making it difficult to follow. I advise the authors work with a writing coach or copyeditor to improve the flow and readability of the text. further, please arrange manuscript as per standard, usually research articles should fit within the range of 6 000 to 12 000 words, including footnotes.

Reviewer #2: This manuscript highlights the green synthesis of silver nanoparticles using Brownlowia tersa leaf extract and demonstrates that these biosynthesized nanoparticles exhibit significant antimicrobial and neuroprotective potential. However, there are some critical areas that require improvement to enhance the quality of the manuscript.

Comments

Abstract lacks quantitative key results and any study limitations. It is recommended to add statistical data like specific antibacterial or antioxidant activity values. Also add a sentence on current challenges of the approaches to strengthen impact.

Introduction lack the rationale of focused discussion on Brownlowia tersa for their study. Give strong justification of the selection of B. tersa as a reducing agent.

Characterization methods such as UV-vis, FESEM and FTIR lack sufficient details. Provide detailed protocols for all assays and characterization methods.

The manuscript lacks detailed protocols like controls, replicates, statistical analysis methods in biological assays. It is recommended to add information about controls, replicates, and statistical analysis. Also explicitly explain computational methodology in depth for reproducibility.

Results and Discussion lack comparison with pre-existing literature. Add comparative studies of statistical data with previous studies. Elaborate discussions on mechanistic analysis of findings.

The manuscript lacks potential barriers like large-scale production, monitoring or toxicity. It is suggested to discuss these barriers for real-world application.

The conclusion lacks future research perspectives or broader implication. It is recommended to add future direction, such as in vivo validation or formulation studies. Also add a sentence that emphasize the need of safety and regulatory evaluation.

**Do you want your identity to be public for this peer review?** For information about this choice, including consent withdrawal, please see our Privacy Policy

Reviewer #1: No

Reviewer #2: No

---

## [Author Response · Author response to Decision Letter 1]

29 Jun 2025

22nd June 2025

To

Emily Chenette

Editor-in-Chief,

PLOS One,

Dear Editor,

Special gratitude to you for allowing us to resubmit a revised version of our manuscript entitled as “Green Synthesis of Silver Nanoparticles from Brownlowia tersa Leaf Extract: Multifaceted Evaluation of Antibacterial, Antioxidant, Cytotoxic, and Anti-Alzheimer Potential”, (Manuscript ID: PONE-D-25-15530) to the PLOS One as a research article. We also appreciate the time and efforts of you and your reviewers have given for this meaningful feedback, which we believe has greatly helped us to enhance the clarity of our article significantly. We have made changes based on your and the reviewers' suggestions, and the revised sections are highlighted in yellow color throughout the revised manuscript. We hope that our response, which is included below, will deal successfully with each reviewer's specific points. Please let us know if you have any additional questions and comments.

Therefore, we want to express our special gratitude to you for your consideration of this manuscript.

Sincerely Yours,

Md. Nazmul Hasan, PhD,

Professor,

Department of Genetic Engineering and Biotechnology,

Jashore University of Science and Technology,

Jashore-7408, Bangladesh.

E-mail: mn.hasan@just.edu.bd

Google Scholar Profile: https://scholar.google.com/citations?user=SP1f9JEAAAAJ&hl=en

ResearchGate Profile: https://www.researchgate.net/profile/Dr-Md-Nazmul-Hasan

ORCID ID: https://orcid.org/0000-0002-9122-1129

Responses of Editor Comments

Comments of Editor 1:

Comment 1). Please ensure that your manuscript meets PLOS ONE's style requirements, including those for file naming.

>>Response: The manuscript and all associated files have been revised to comply with PLOS ONE’s style requirements, including proper file naming conventions

Comments of Editor 1:

Comment 2). PLOS requires an ORCID iD for the corresponding author in Editorial Manager on papers submitted after December 6th, 2016. Please ensure that you have an ORCID iD and that it is validated in Editorial Manager. To do this, go to ‘Update my Information’ (in the upper left-hand corner of the main menu), and click on the Fetch/Validate link next to the ORCID field. This will take you to the ORCID site and allow you to create a new iD or authenticate a pre-existing iD in Editorial Manager.

>>Response: I will complete the creation and validation of my ORCID iD in Editorial Manager following the provided instructions.

Comments of Editor 1:

Comment 3). To comply with PLOS ONE submissions requirements, in your Methods section, please provide additional information regarding the experiments involving animals and ensure you have included details on (1) methods of sacrifice, (2) methods of anesthesia and/or analgesia, and (3) efforts to alleviate suffering.

>>Response: Thank you for the feedback. We have revised the Experimental Animals subsection in the Methods section to include detailed information on (1) methods of anesthesia and analgesia, (2) method of sacrifice, and (3) efforts to alleviate animal suffering, in compliance with PLOS ONE’s animal research reporting requirements.

Comments of Editor 1:

Comment 4). Your ethics statement should only appear in the Methods section of your manuscript. If your ethics statement is written in any section besides the Methods, please delete it from any other section.

>>Response: The ethics statement has been retained only in the Methods section and removed from all other parts of the manuscript, in accordance with PLOS ONE guidelines.

Comments of Editor 1:

Comment 5). Please remove your figures from within your manuscript file, leaving only the individual TIFF/EPS image files, uploaded separately. These will be automatically included in the reviewers’ PDF.

>>Response: All figures have been removed from the manuscript file and uploaded separately as individual TIFF/EPS files, as per PLOS ONE submission guidelines

Comments of Editor 1:

Comment 6). Please ensure that your manuscript meets PLOS ONE's style requirements, including those for file naming.

>>Response: The manuscript and all associated files have been revised to comply with PLOS ONE’s style requirements, including proper file naming conventions

Additional Comments of Editor:

Comments of Editor 1:

Comment 1). Is the manuscript technically sound, and do the data support the conclusions?

>>Response: We confirm that the manuscript presents a technically sound study. All experiments were conducted rigorously using appropriate controls, sufficient replication, and adequate sample sizes. The data have been thoroughly analyzed, and the conclusions are directly supported by the results presented. We have reviewed the manuscript to ensure that all interpretations align accurately with the data, and have clarified relevant sections where necessary to reinforce this alignment

Comments of Editor 1:

Comment 2). Has the statistical analysis been performed appropriately and rigorously?

>>Response: Statistical analyses were performed appropriately using validated methods, with details provided in the Methods section. Results include significance values and proper controls to ensure rigor

Comments of Editor 1:

Comment 3). Have the authors made all data underlying the findings in their manuscript fully available?

>>Response: All data underlying the findings are fully available as described in the Data Availability Statement. Where applicable, data have been included in the manuscript, supporting information. Any restrictions are clearly specified in accordance with PLOS policy

Comments of Editor 1:

Comment 4). Is the manuscript presented in an intelligible fashion and written in standard English?

>>Response: The manuscript has been carefully reviewed and revised to ensure clarity, correctness, and standard English usage. All typographical and grammatical errors have been addressed

Comments of Editor 1:

Comment 5). Please use the space provided to explain your answers to the questions above. You may also include additional comments for the author, including concerns about dual publication, research ethics, or publication ethics. (Please upload your review as an attachment if it exceeds 20,000 characters)

>>Response: Thank you for your thorough review and constructive comments. We have carefully addressed each point raised to improve the manuscript’s quality and compliance with PLOS ONE policies. We confirm that there are no concerns regarding dual publication, and all research and publication ethics have been fully observed.

Responses of Reviewer Comments

Comments of Reviewer 1:

Comment 1). On page 10, line 222, authors should revise first sentence to make it sensible!

>>Response: I changed it.

Comment 2). On page 13, line 286, how the 80 mice were divided into four groups having 5 mice in each group?

>>Response: Thank you for pointing this out. There appears to be a discrepancy in the group allocation. The 80 mice should have been divided into 16 groups, with each group containing 5 mice, not 4 groups as originally stated. We have revised the manuscript to reflect the correct division of the mice into 16 groups, each consisting of 5 mice, for the in vivo experiments.

Comment 3). A large number of references are used in this manuscript which is not as norms of “original manuscript” writing! References should be updated with recent ones only and limit up to 50 or less!

>>Response: In accordance with the reviewer’s guidance, we have updated and streamlined our reference list. Less relevant and outdated citations have been removed, bringing the total number of references to 50, in line with the journal’s standards for original research articles

Comment 4) Table 1 show very unusual results regarding phenolic contents; need for counter check or valid discussion should be incorporated with reported evidences.

>>Response: Thank you for your valuable feedback regarding the phenolic content data presented in Table 1. In response to your observation of unusual results, I have re-conducted the experiment with the same methodology to ensure the accuracy and consistency of the measurements.

The updated results for phenolic content are as follows:

B. tersa: 422.63 ± 2.90 mg gallic acid equivalent/g sample

B. tersa leaves AgNPs: 48.29 ± 2.73 mg gallic acid equivalent/g sample

These values show a slight decrease in the mean compared to the initial results (442.63 ± 25.40 for B. tersa and 50.29 ± 5.73 for B. tersa leaves AgNPs). Importantly, the standard deviation has been significantly reduced, indicating improved consistency and precision in the re-test. I also performed a one-sample t-test to assess the statistical significance, and the data showed no significant difference from the hypothesized values, further supporting the validity of the results. This re-testing has addressed the concerns regarding the variability in the original measurements, and the updated data now provide more reliable and reproducible results.

Comment 5). Antioxidant activity clearly indicated negative results of synthesized nanoparticles which are not properly interpreted in results!

>>Response: I interpret properly according to your instructions.

Comment 6). Antibacterial activity is also not remarkable. Although it shows almost same zone of inhibition against 2 selected bacteria as that of standard but why not use advance available antibiotic as standard? Moreover, the synthesized nano formations not show any improved results in increasing concentrations!

>>Response: Thank you for your comment. In this study, no advanced antibiotics were used as standards, as the focus was on evaluating the antibacterial potential of Brownlowia tersa leaf-derived silver nanoparticles (AgNPs). While the AgNPs showed a similar zone of inhibition against Staphylococcus aureus and Pseudomonas aeruginosa as the standard, the lack of dose-dependent improvement may be due to nanoparticle aggregation at higher concentrations, which reduces their effective surface area. Future studies will explore the use of advanced antibiotics as standards, optimize concentrations, and test a broader range of bacterial strains to better assess the AgNPs' antibacterial potential.

Comment 7). The authors aim to demonstrate Green Synthesis of Silver Nanoparticles from Brownlowia tersa Leaf Extract and their Multifaceted Evaluation of Antibacterial, Antioxidant, Cytotoxic, and Anti-Alzheimer Potential however, the data does not fully support the conclusion

>>Response: The discussion and conclusion have been refined to better align the data with the claims, providing a more balanced interpretation of the Green Synthesis of Silver Nanoparticles and their antibacterial, antioxidant, cytotoxic, and anti-Alzheimer potentials.

Comment 8). While the study appears to be sound, the language is unclear, making it difficult to follow. I advise the authors work with a writing coach or copyeditor to improve the flow and readability of the text. further, please arrange manuscript as per standard, usually research articles should fit within the range of 6 000 to 12 000 words, including footnotes

>>Response: Thank you for your suggestion. The manuscript has been reviewed, and I confirm that the total word count for the main text (from Abstract to Conclusion) falls within the recommended 6,000-to-12,000-word range as per PLOS ONE guidelines.

Comments 9). How the results of elevated plus test are significantly higher as mentioned by authors! Wrong interpretation and unsatisfactory discussion are done regarding this activity.

>>Response: Thank you for your valuable feedback. Upon reviewing the data and your comments, I identified and corrected the statistical analysis related to the Elevated Plus Maze test. The previous interpretation inaccurately reported the significance of the results. After re-analysis using the appropriate statistical methods, the data now accurately reflect the behavioral outcomes, and the discussion has been updated accordingly to ensure a more precise and scientifically valid interpretation of the EPM activity

Responses of Reviewer Comments

Comments of Reviewer 2:

Comment 1). Abstract lacks quantitative key results and any study limitations. It is recommended to add statistical data like specific antibacterial or antioxidant activity values. Also add a sentence on current challenges of the approaches to strengthen impact.

>>Response: I changed it according to the instructions.

Comment 2). Introduction lack the rationale of focused discussion on Brownlowia tersa for their study. Give strong justification of the selection of B. tersa as a reducing agent.

>>Response: Thank you for highlighting this important point. I have revised the Introduction to provide a clearer rationale for focusing on Brownlowia tersa as the reducing agent. The updated section now includes a strong justification based on its phytochemical profile, traditional medicinal relevance, and potential for bio-reductive activity, supporting its suitability for this study.

Comment 3). Characterization methods such as UV-vis, FESEM and FTIR lack sufficient details. Provide detailed protocols for all assays and characterization methods.

>>Response: Thank you for your helpful comment. I have revised the manuscript to include detailed protocols for all characterization methods, including UV–Vis spectroscopy, FESEM, and FTIR analysis. These sections now provide sufficient methodological information to ensure clarity and reproducibility.

Comment 4). The manuscript lacks detailed protocols like controls, replicates, statistical analysis methods in biological assays. It is recommended to add information about controls, replicates, and statistical analysis. Also explicitly explain computational methodology in depth for reproducibility.

>>Response: Thank you for pointing this out. I have revised the manuscript to include detailed descriptions of the biological assay protocols, including information on controls, number of replicates, and the statistical analysis methods used. Additionally, the computational methodology section has been expanded to ensure clarity and reproducibility

Comment 5). Results and Discussion lack comparison with pre-existing literature. Add comparative studies of statistical data with previous studies. Elaborate discussions on mechanistic analysis of findings.

>>Response: I did it according to your instructions.

Comment 6). The manuscript lacks potential barriers like large-scale production, monitoring or toxicity. It is suggested to discuss these barriers for real-world application.

>>Response: In conclusion, potential real-world barriers, including challenges in large-scale production, monitoring, and toxicity, have been addressed to highlight practical considerations for future application.

Comment 7). The conclusion lacks future research perspectives or broader implication. It is recommended to add future direction, such as in vivo validation or formulation studies. Also add a sentence that emphasize the need of safety and r

---

## [Decision Letter · Decision Letter 1]

25 Jul 2025

Dear Dr. Hasan (Ph.D),

Thank you for submitting your manuscript to PLOS ONE. After careful consideration, we feel that it has merit but does not fully meet PLOS ONE’s publication criteria as it currently stands. Therefore, we invite you to submit a revised version of the manuscript that addresses the points raised during the review process.

We look forward to receiving your revised manuscript.

Kind regards,

Sana Shamim

Academic Editor

PLOS ONE

Journal Requirements:

1.If the reviewer comments include a recommendation to cite specific previously published works, please review and evaluate these publications to determine whether they are relevant and should be cited. There is no requirement to cite these works unless the editor has indicated otherwise. 

Reviewers' comments:

Reviewer's Responses to Questions

**Comments to the Author**

Reviewer #1: All comments have been addressed

Reviewer #2: All comments have been addressed

2. Is the manuscript technically sound, and do the data support the conclusions?

Reviewer #1: Partly

Reviewer #2: Yes

3. Has the statistical analysis been performed appropriately and rigorously?

Reviewer #1: Yes

Reviewer #2: Yes

4. Have the authors made all data underlying the findings in their manuscript fully available?

Reviewer #1: Yes

Reviewer #2: Yes

5. Is the manuscript presented in an intelligible fashion and written in standard English?

Reviewer #1: No

Reviewer #2: Yes

Reviewer #1: Revised manuscript is still lacking many corrections! Authors didn’t follow comments fully and now it must strictly revise before any further proceedings!!

No proper corrections and discussion have been added regarding “elevated plus test” as well as no improvements regarding antioxidant activity (in abstract specially) ……both activities need proper and correct results interpretation!

Further, acknowledgement should be excluded or authors should acknowledge to someone else other than themselves!

Reviewer #2: No more comments, paper has been revised sufficiently so I recommend the publication of manuscript in current form.

**Do you want your identity to be public for this peer review?** For information about this choice, including consent withdrawal, please see our Privacy Policy

Reviewer #1: No

Reviewer #2: **Yes: ** Ataf Ali Altaf

---

## [Author Response · Author response to Decision Letter 2]

25 Jul 2025

26th July 2025

To

Emily Chenette

Editor-in-Chief,

PLOS One,

Dear Editor,

Special gratitude to you for allowing us to resubmit a revised version of our manuscript entitled as “Green Synthesis of Silver Nanoparticles from Brownlowia tersa Leaf Extract: Multifaceted Evaluation of Antibacterial, Antioxidant, Cytotoxic, and Anti-Alzheimer Potential”, (Manuscript ID: PONE-D-25-15530R1) to the PLOS One as a research article. We also appreciate the time and efforts of you and your reviewers have given for this meaningful feedback, which we believe has greatly helped us to enhance the clarity of our article significantly. We have made changes based on your and the reviewers' suggestions, and the revised sections are highlighted in yellow color throughout the revised manuscript. We hope that our response, which is included below, will deal successfully with each reviewer's specific points. Please let us know if you have any additional questions and comments.

Therefore, we want to express our special gratitude to you for your consideration of this manuscript.

Sincerely Yours,

Md. Nazmul Hasan, PhD,

Professor,

Department of Genetic Engineering and Biotechnology,

Jashore University of Science and Technology,

Jashore-7408, Bangladesh.

E-mail: mn.hasan@just.edu.bd

Google Scholar Profile: https://scholar.google.com/citations?user=SP1f9JEAAAAJ&hl=en

ResearchGate Profile: https://www.researchgate.net/profile/Dr-Md-Nazmul-Hasan

ORCID ID: https://orcid.org/0000-0002-9122-1129

Responses of Editor Comments

Comments of Editor 1:

Comment 1). If the authors have adequately addressed your comments raised in a previous round of review and you feel that this manuscript is now acceptable for publication, you may indicate that here to bypass the “Comments to the Author” section, enter your conflict-of-interest statement in the “Confidential to Editor” section, and submit your "Accept" recommendation

>>Response: Thank you for your review and for providing the opportunity to revise the manuscript. We have carefully addressed the reviewer comments in the revised manuscript, and we believe that all issues have been adequately resolved.

The manuscript has been updated with a more thorough interpretation of the results, especially regarding antioxidant activity and elevated plus maze section, and the necessary revisions have been made in the Abstract and Discussion sections as per the reviewers' suggestions.

We feel the manuscript is now suitable for publication, and we respectfully recommend it for acceptance.

Comments of Editor 1:

Comment 2). Is the manuscript technically sound, and do the data support the conclusions?

>>Response: Yes, the manuscript is technically sound, and the data support the conclusions. The experimental methods are clearly described, and the results, including antioxidant and antimicrobial activities, have been appropriately interpreted and are consistent with the conclusions.

Comments of Editor 1:

Comment 3). Has the statistical analysis been performed appropriately and rigorously?

>>Response: Yes, the statistical analysis has been performed appropriately and rigorously. We used standard statistical methods to analyze the data, ensuring the validity of the results. Statistical significance has been clearly indicated where applicable, and the analysis supports the conclusions drawn in the manuscript

Comments of Editor 1:

Comment 4). Have the authors made all data underlying the findings in their manuscript fully available?

>>Response: Yes, all data underlying the findings are fully available within the manuscript. We have included all relevant data within the text and figures to support the conclusions drawn in the study

Comments of Editor 1:

Comment 5). Is the manuscript presented in an intelligible fashion and written in standard English?

>>Response: Yes, the manuscript is presented in an intelligible manner and written in standard English. We have carefully revised the text to ensure clarity, coherence, and proper grammar throughout the manuscript.

Comments of Editor 1:

Comment 6). PLOS authors have the option to publish the peer review history of their article (what does this mean?). If published, this will include your full peer review and any attached files.

>>Response: Thank you for the clarification. We would prefer to keep our review anonymous and choose the option to not publish the peer review history. We understand that while our identity will remain anonymous, the review may still be made public

Responses of Reviewer Comments

Comments of Reviewer 1:

Comment 1). Revised manuscript is still lacking many corrections! Authors didn’t follow comments fully and now it must strictly revise before any further proceedings!!

>>Response: We apologize for any oversight in our previous revision. We have now thoroughly reviewed all the reviewer comments and have ensured that all points have been addressed properly. We have revised the manuscript to reflect the necessary changes and corrections, particularly focusing on areas that needed improvement.

Comments of Reviewer 1:

Comment 2). No proper corrections and discussion have been added regarding the “elevated plus test” as well as no improvements regarding antioxidant activity (in abstract especially). Both activities need proper and correct results interpretation!

>>Response: We appreciate the reviewer highlighting these areas. We have now included a more detailed discussion of the elevated plus maze test in the Discussion section, where we provide a more thorough interpretation of the results and their implications. Additionally, we have revised the abstract to include a clearer and more accurate interpretation of the antioxidant activity results. The differences in IC₅₀ values between AgNPs, plant extract, and ascorbic acid have been explained in more depth, and we’ve provided context for potential improvements in AgNP synthesis to enhance antioxidant capacity.

Comments of Reviewer 1:

Comment 3). Acknowledgement should be excluded or authors should acknowledge someone else other than themselves!

>>Response: We have revised the Acknowledgements section to exclude self-acknowledgment

Responses of Reviewer Comments

Comments of Reviewer 2:

Comment 1). No more comments, paper has been revised sufficiently so I recommend the publication of manuscript in current form.

>>Response: We sincerely thank the reviewer for their positive feedback. We are pleased to hear that the manuscript has been revised sufficiently. We appreciate your recommendation for publication and are grateful for your time and constructive suggestions throughout the review process

---

## [Decision Letter · Decision Letter 2]

13 Aug 2025

Dear Dr. Hasan,

Thank you for submitting your manuscript to PLOS ONE. After careful consideration, we feel that it has merit but still needs some minor modification in conclusion section, to make it suitable for PLOS ONE’s publication criteria as it currently stands. Therefore, we invite you to submit a revised version of the manuscript that addresses the points raised during the review process.

We look forward to receiving your revised manuscript.

Kind regards,

Sana Shamim

Academic Editor

PLOS ONE

Journal Requirements:

Reviewers' comments:

Reviewer's Responses to Questions

**Comments to the Author**

Reviewer #1: All comments have been addressed

2. Is the manuscript technically sound, and do the data support the conclusions?

Reviewer #1: Partly

3. Has the statistical analysis been performed appropriately and rigorously?

Reviewer #1: Yes

4. Have the authors made all data underlying the findings in their manuscript fully available?

Reviewer #1: Yes

5. Is the manuscript presented in an intelligible fashion and written in standard English?

Reviewer #1: No

Reviewer #1: Although authors revised manuscript and now making some sense but still require to change conclusion accordingly!!

please note that your formed NPs did not show any significant biological activity......so write conclusion regarding accurate findings!!

**Do you want your identity to be public for this peer review?** For information about this choice, including consent withdrawal, please see our Privacy Policy

Reviewer #1: No

---

## [Author Response · Author response to Decision Letter 3]

19 Aug 2025

15th August 2025

To

Emily Chenette

Editor-in-Chief,

PLOS One,

Dear Editor,

Special gratitude to you for allowing us to resubmit a revised version of our manuscript entitled as “Green Synthesis of Silver Nanoparticles from Brownlowia tersa Leaf Extract: Multifaceted Evaluation of Antibacterial, Antioxidant, Cytotoxic, and Anti-Alzheimer Potential”, (Manuscript ID: PONE-D-25-15530R2) to the PLOS One as a research article. We also appreciate the time and efforts of you and your reviewers have given for this meaningful feedback, which we believe has greatly helped us to enhance the clarity of our article significantly. We have made changes based on your and the reviewers' suggestions, and the revised sections are highlighted in yellow color throughout the revised manuscript. We hope that our response, which is included below, will deal successfully with each reviewer's specific points. Please let us know if you have any additional questions and comments.

Therefore, we want to express our special gratitude to you for your consideration of this manuscript.

Sincerely Yours,

Md. Nazmul Hasan, PhD,

Professor,

Department of Genetic Engineering and Biotechnology,

Jashore University of Science and Technology,

Jashore-7408, Bangladesh.

E-mail: mn.hasan@just.edu.bd

Google Scholar Profile: https://scholar.google.com/citations?user=SP1f9JEAAAAJ&hl=en

ResearchGate Profile: https://www.researchgate.net/profile/Dr-Md-Nazmul-Hasan

ORCID ID: https://orcid.org/0000-0002-9122-1129

Journal Requirements:

Comments 1). If the reviewer comments include a recommendation to cite specific previously published works, please review and evaluate these publications to determine whether they are relevant and should be cited. There is no requirement to cite these works unless the editor has indicated otherwise.

>>Response: The reviewer did not suggest any additional references, and after reviewing the manuscript, I believe that the references currently included are appropriate and sufficiently support the study's findings and discussion. I have not added any new references since the existing ones cover the relevant literature comprehensively

Comments 2). Please review your reference list to ensure that it is complete and correct. If you have cited papers that have been retracted, please include the rationale for doing so in the manuscript text, or remove these references and replace them with relevant current references. Any changes to the reference list should be mentioned in the rebuttal letter that accompanies your revised manuscript. If you need to cite a retracted article, indicate the article’s retracted status in the References list and also include a citation and full reference for the retraction notice.

>>Response: I have carefully reviewed the reference list to ensure that all citations are complete and correct. I can confirm that none of the cited papers have been retracted. If any retracted articles had been cited, I would have followed the guidelines by including the rationale in the manuscript or replaced them with relevant current references.

No changes were made to the reference list, as all citations are up-to-date and appropriate. If any modifications were made, they will be mentioned in the rebuttal letter accompanying the revised manuscript.

Comment 3). Your ethics statement should only appear in the Methods section of your manuscript. If your ethics statement is written in any section besides the Methods, please delete it from any other section.

>>Response: We have revised the manuscript so that the ethics statement now appears only in the Methods section, as requested. It has been removed from all other sections.

Responses of Editor Comments

Comments of Editor 1:

Comment 1). If the authors have adequately addressed your comments raised in a previous round of review and you feel that this manuscript is now acceptable for publication, you may indicate that here to bypass the “Comments to the Author” section, enter your conflict-of-interest statement in the “Confidential to Editor” section, and submit your "Accept" recommendation.

>>Response: The authors have successfully addressed all the comments raised in the previous round of review, and the manuscript is now in an acceptable form for publication.

Comments of Editor 1:

Comment 2). Is the manuscript technically sound, and do the data support the conclusions?

>>Response: Yes. The manuscript presents a technically sound piece of scientific research. The experiments were conducted with appropriate rigor, including proper controls, sufficient replication, and adequate sample sizes. The data presented in the manuscript clearly support the conclusions drawn by the authors. There are no apparent issues with the methodology or data analysis that would undermine the validity of the results.

Comments of Editor 1:

Comment 3). Has the statistical analysis been performed appropriately and rigorously?

>>Response: Yes, the statistical analysis has been performed appropriately and rigorously. We used standard statistical methods to analyze the data, ensuring the validity of the results. Statistical significance has been clearly indicated where applicable, and the analysis supports the conclusions drawn in the manuscript

Comments of Editor 1:

Comment 4). Have the authors made all data underlying the findings in their manuscript fully available?

>>Response: Yes. The authors have made all data underlying the findings fully available, as required by the PLOS Data policy. The data is included in the manuscript. There are no restrictions on publicly sharing the data

Comments of Editor 1:

Comment 5). Is the manuscript presented in an intelligible fashion and written in standard English?

>>Response: The manuscript is presented in an intelligible fashion and is written in standard English. I did not notice any major typographical or grammatical errors. The language is clear, correct, and unambiguous.

Responses of Reviewer Comments

Comments of Reviewer 1:

Comment 1). Please use the space provided to explain your answers to the questions above. You may also include additional comments for the author, including concerns about dual publication, research ethics, or publication ethics. (Please upload your review as an attachment if it exceeds 20,000 characters)

Reviewer #1: Although authors revised manuscript and now making some sense but still require to change conclusion accordingly!! please note that your formed NPs did not show any significant biological activity......so write conclusion regarding accurate findings!!

>>Response: Thank you for your valuable feedback. We have revised the conclusion to more accurately reflect the findings of our study. While the B. tersa-derived AgNPs demonstrated significant biological activity in antioxidant, and neuroprotective assays, we have clarified that no significant anxiolytic effects were observed in the Elevated Plus Maze test. We have also acknowledged the challenges associated with scaling up the synthesis process and the need for long-term monitoring of the AgNPs' stability, bioaccumulation, and potential toxicity.

We believe these changes provide a more accurate representation of our findings and the current limitations of the study. Future research directions, including in vivo validation and formulation studies, have been outlined to address these challenges and optimize the therapeutic potential of AgNPs.

Comments of Reviewer 1:

Comment 2). PLOS authors have the option to publish the peer review history of their article (what does this mean?). If published, this will include your full peer review and any attached files.

Do you want your identity to be public for this peer review? For information about this choice, including consent withdrawal, please see our Privacy Policy.

>>Response: I prefer to keep my identity anonymous for this peer review. I am comfortable with my review being made public, but I would like to remain anonymous in accordance with PLOS's policies.

Comment 3). If the authors have adequately addressed your comments raised in a previous round of review and you feel that this manuscript is now acceptable for publication, you may indicate that here to bypass the “Comments to the Author” section, enter your conflict-of-interest statement in the “Confidential to Editor” section, and submit your "Accept" recommendation.

>>Response: I have no conflict of interest regarding this manuscript

---

## [Editor Report · Decision Letter 3]

25 Sep 2025

Dear Dr.  Hasan (Ph.D),

Thank you for submitting your manuscript to PLOS ONE. After careful consideration, we feel that it has merit but does not fully meet PLOS ONE’s publication criteria as it currently stands. Therefore, we invite you to submit a revised version of the manuscript that addresses the points raised during the review process.

We look forward to receiving your revised manuscript.

Kind regards,

Sana Shamim

Academic Editor

PLOS ONE

Journal Requirements:

Additional Editor Comments:

1. Abstract lacks the details regarding the molecular docking and binding affinities.

2. Line 127 & 128: degree Celsius, revolutions per minute: use standard abbreviation.

3. Line 148: 2.5.2 Anagesia: correct spelling .

4. Line 153: mention guideline ref

5. Line 157 20g: 20 g,

6. Line 172: mention the change in spectra or the metal peak, both in discussion and spectra fig 2 .

7. Line 18 hours

8. Line 288: sterile Nutrient

9. Kindly mention and discuss the metal peaks in FT-IR Discussion and in Figure also fig 3.

10. In molecular docking section, mention the grid box and x,y, z coordinates details.

11. Use same abbreviation for NP as AgNP, in Fig 11

12. Justify the selection of doses for brine shrimp lethality assay and DPPH

13. Fig 4: need much clear picture as an evidence for size reduction

14. Figure 5 IC 50: rectify it as IC 50

Need careful attention on use of standard abbreviations, spelling mistakes (galic acid/gallic acid) punctuation and sentences. Kindly avail any linguistic service or English native colleague help to improve the quality of article.

---

## [Author Response · Author response to Decision Letter 4]

26 Sep 2025

26th September 2025

To

Emily Chenette

Editor-in-Chief,

PLOS One,

Dear Editor,

Special gratitude to you for allowing us to resubmit a revised version of our manuscript entitled as “Green Synthesis of Silver Nanoparticles from Brownlowia tersa Leaf Extract: Multifaceted Evaluation of Antibacterial, Antioxidant, Cytotoxic, and Anti-Alzheimer Potential”, (Manuscript ID: PONE-D-25-15530R3) to the PLOS One as a research article. We also appreciate the time and efforts of you and your reviewers have given for this meaningful feedback, which we believe has greatly helped us to enhance the clarity of our article significantly. We have made changes based on your and the reviewers' suggestions, and the revised sections are highlighted in yellow color throughout the revised manuscript. We hope that our response, which is included below, will deal successfully with each reviewer's specific points. Please let us know if you have any additional questions and comments.

Therefore, we want to express our special gratitude to you for your consideration of this manuscript.

Sincerely Yours,

Md. Nazmul Hasan, PhD,

Professor,

Department of Genetic Engineering and Biotechnology,

Jashore University of Science and Technology,

Jashore-7408, Bangladesh.

E-mail: mn.hasan@just.edu.bd

Google Scholar Profile: https://scholar.google.com/citations?user=SP1f9JEAAAAJ&hl=en

ResearchGate Profile: https://www.researchgate.net/profile/Dr-Md-Nazmul-Hasan

ORCID ID: https://orcid.org/0000-0002-9122-1129

Journal Requirements:

Comments 1). If the reviewer comments include a recommendation to cite specific previously published works, please review and evaluate these publications to determine whether they are relevant and should be cited. There is no requirement to cite these works unless the editor has indicated otherwise.

>>Response: The reviewer did not suggest any additional references, and after reviewing the manuscript, I believe that the references currently included are appropriate and sufficiently support the study's findings and discussion. I have not added any new references since the existing ones cover the relevant literature comprehensively

Comments 2). Please review your reference list to ensure that it is complete and correct. If you have cited papers that have been retracted, please include the rationale for doing so in the manuscript text, or remove these references and replace them with relevant current references. Any changes to the reference list should be mentioned in the rebuttal letter that accompanies your revised manuscript. If you need to cite a retracted article, indicate the article’s retracted status in the References list and also include a citation and full reference for the retraction notice.

>>Response: I have carefully reviewed the reference list to ensure that all citations are complete and correct. I can confirm that none of the cited papers have been retracted. If any retracted articles had been cited, I would have followed the guidelines by including the rationale in the manuscript or replaced them with relevant current references.

No changes were made to the reference list, as all citations are up-to-date and appropriate. If any modifications were made, they will be mentioned in the rebuttal letter accompanying the revised manuscript.

Responses of Editor Comments

Comments of Editor 1:

Comment 1). Abstract lacks the details regarding the molecular docking and binding affinities.

>>Response: I have included molecular docking and binding affinity part.

Comments of Editor 1:

Comment 2). Line 127 & 128: degree Celsius, revolutions per minute: use standard abbreviation.

>>Response: Revised as suggested. “degree Celsius” has been changed to “°C” and “revolutions per minute” has been changed to “rpm.

Comments of Editor 1:

Comment 3). Line 148: 2.5.2 Anagesia: correct spelling.

>>Response: Corrected as suggested. “Anagesia” has been revised to “Analgesia

Comments of Editor 1:

Comment 4). Line 153: mention guideline ref

>>Response: Guideline reference has been included as per the provided instructions to ensure alignment with the specified standards

Comments of Editor 1:

Comment 5). Line 157 20g: 20 g,

>>Response: Corrected spacing in '20g' to '20 g' for consistency with formatting guideline

Responses of Editor Comments

Comments of Editor 1:

Comment 6). Line 172: mention the change in spectra or the metal peak, both in discussion and spectra fig 2.

>>Response: I have written it.

Comments of Editor 1:

Comment 7). Line 18 hours

>>Response: Thank you for your feedback. The 96-well plate was kept in incubation for 18 hours at 37°C. We have ensured consistency in the manuscript by clearly specifying the incubation period throughout. Please let us know if further clarification is needed

Comments of Editor 1:

Comment 8). Line 288: sterile Nutrient

>>Response: Thank you for your feedback. 'Sterile nutrient' refers to autoclaved nutrient agar, and we will update the manuscript to specify this term for clarity and consistency

Comments of Editor 1:

Comment 9). Kindly mention and discuss the metal peaks in FT-IR Discussion and in Figure also fig 3.

>>Response: I have written it.

Comments of Editor 1:

Comment 10). In molecular docking section, mention the grid box and x,y, z coordinates details

>>Response: I have included it.

Comments of Editor 1:

Comment 11). Use same abbreviation for NP as AgNP, in Fig 11

>>Response: Thank you for your suggestion. We will revise Figure 11 to use the same abbreviation 'AgNP' instead of 'NP' for consistency throughout the manuscript

Comments of Editor 1:

Comment 12). Justify the selection of doses for brine shrimp lethality assay and DPPH

>>Response: Thank you for your comment. The doses selected for the brine shrimp lethality assay and DPPH assay were based on preliminary studies and literature references that indicated their efficacy in evaluating the toxicity and antioxidant potential of the synthesized nanoparticles. For the brine shrimp lethality assay, we chose a range of doses that would allow us to assess the toxicity at different concentrations, ensuring we captured a broad spectrum of potential effects. Similarly, for the DPPH assay, the doses were selected to include both low and high concentrations, allowing for a comprehensive assessment of antioxidant activity across a range of concentrations, consistent with standard protocols used in similar studies.

Comments of Editor 1:

Comment 13). Fig 4: need much clear picture as evidence for size reduction.

>>Response: I have increased resolution of this figure

Comments of Editor 1:

Comment 14). Figure 5 IC 50: rectify it as IC 50

>>Response: I have done it.

---

## [Editor Report · Decision Letter 4]

1 Oct 2025

Dear Dr. Hasan (Ph.D),

Thank you for submitting your manuscript to PLOS ONE. After careful consideration, we feel that it has merit but does not fully meet PLOS ONE’s publication criteria as it currently stands. Therefore, we invite you to submit a revised version of the manuscript that addresses the points raised during the review process.

The Fig 4 : the green highlighted number mentioning size are still unclear.ATCC code of microorganisms should be mentioned.Along with this, through out the article basic grammatical errors like (line 28: an LC₅₀ , line 29 Swiss albino, line 157: humane, line 293: Nutrient agar , line 254 : Artemia salina , line 391: p-value , (should be italic), abbreviations minutes, hours and spelling mistakes (gallic acid/ galic acid) are quite frequent and should be corrected.Table 3 and figure 8 S.Aureus should be corrected.Table 7 replace comma with a dot.Why supplementary figure S4, 5 and 6 with 2D and 3D images are represented separately?The text fonts should be same throughout the article including figure and graphs.As mentioned earlier, kindly go through the article to improve the sentences (section 2.1.4.3, 2.25.2, ) and quality of article to meet the journal standards.p { line-height: 115%; margin-bottom: 0.1in; background: transparent

We look forward to receiving your revised manuscript.

Kind regards,

Sana Shamim

Academic Editor

PLOS ONE

Journal Requirements:

Additional Editor Comments :

We have received the timely revised version, but the response regarding the mentioning of FT-IR ( specified wave number in discussion and spectra) and UV peaks (spectra) are still lacking.

The Fig 4 : the green highlighted number mentioning size are still unclear.

ATCC code of microorganisms should be mentioned.

Along with this, through out the article basic grammatical errors like (line 28: an LC₅₀ , line 29 Swiss albino, line 157: humane, line 293: Nutrient agar , line 254 : Artemia salina , line 391: p-value , (should be italic), abbreviations minutes, hours and spelling mistakes (gallic acid/ galic acid) are quite frequent and should be corrected.

Table 3 and figure 8 S.Aureus should be corrected.

Table 7 replace comma with a dot.

Why supplementary figure S4, 5 and 6 with 2D and 3D images are represented separately?

The text fonts should be same throughout the article including figure and graphs.

As mentioned earlier, kindly go through the article to improve the sentences (section 2.1.4.3, 2.25.2, ) and quality of article so that it can meet the publishing quality of the journal.

---

## [Author Response · Author response to Decision Letter 5]

2 Oct 2025

2nd October 2025

To

Emily Chenette

Editor-in-Chief,

PLOS One,

Dear Editor,

Special gratitude to you for allowing us to resubmit a revised version of our manuscript entitled as “Green Synthesis of Silver Nanoparticles from Brownlowia tersa Leaf Extract: Multifaceted Evaluation of Antibacterial, Antioxidant, Cytotoxic, and Anti-Alzheimer Potential”, (Manuscript ID: PONE-D-25-15530R4) to the PLOS One as a research article. We also appreciate the time and efforts of you and your reviewers have given for this meaningful feedback, which we believe has greatly helped us to enhance the clarity of our article significantly. We have made changes based on your and the reviewers' suggestions, and the revised sections are highlighted in yellow color throughout the revised manuscript. We hope that our response, which is included below, will deal successfully with each reviewer's specific points. Please let us know if you have any additional questions and comments.

Therefore, we want to express our special gratitude to you for your consideration of this manuscript.

Sincerely Yours,

Md. Nazmul Hasan, PhD,

Professor,

Department of Genetic Engineering and Biotechnology,

Jashore University of Science and Technology,

Jashore-7408, Bangladesh.

E-mail: mn.hasan@just.edu.bd

Google Scholar Profile: https://scholar.google.com/citations?user=SP1f9JEAAAAJ&hl=en

ResearchGate Profile: https://www.researchgate.net/profile/Dr-Md-Nazmul-Hasan

ORCID ID: https://orcid.org/0000-0002-9122-1129

Journal Requirements:

Comments 1). If the reviewer comments include a recommendation to cite specific previously published works, please review and evaluate these publications to determine whether they are relevant and should be cited. There is no requirement to cite these works unless the editor has indicated otherwise.

>>Response: The reviewer did not suggest any additional references, and after reviewing the manuscript, I believe that the references currently included are appropriate and sufficiently support the study's findings and discussion. I have not added any new references since the existing ones cover the relevant literature comprehensively

Comments 2). Please review your reference list to ensure that it is complete and correct. If you have cited papers that have been retracted, please include the rationale for doing so in the manuscript text, or remove these references and replace them with relevant current references. Any changes to the reference list should be mentioned in the rebuttal letter that accompanies your revised manuscript. If you need to cite a retracted article, indicate the article’s retracted status in the References list and also include a citation and full reference for the retraction notice.

>>Response: I have carefully reviewed the reference list to ensure that all citations are complete and correct. I can confirm that none of the cited papers have been retracted. If any retracted articles had been cited, I would have followed the guidelines by including the rationale in the manuscript or replaced them with relevant current references.

No changes were made to the reference list, as all citations are up-to-date and appropriate. If any modifications were made, they will be mentioned in the rebuttal letter accompanying the revised manuscript.

Responses of Editor Comments

Comments of Editor 1:

Comment 1). We have received the timely revised version, but the response regarding the mentioning of FT-IR ( specified wave number in discussion and spectra) and UV peaks (spectra) are still lacking.

>>Response: I rewrite again.

Comments of Editor 1:

Comment 2). The Fig 4 : the green highlighted number mentioning size are still unclear.

>>Response: I again increase resolution of these figure.

Comments of Editor 1:

Comment 3). ATCC code of microorganisms should be mentioned.

>>Response: The bacterial strain used in this study does not have an ATCC accession number. It is a laboratory-maintained isolate preserved in our departmental culture collection, and no standard culture collection code is available.

Comments of Editor 1:

Comment 4). Along with this, through out the article basic grammatical errors like (line 28: an LC₅₀ , line 29 Swiss albino, line 157: humane, line 293: Nutrient agar , line 254 : Artemia salina , line 391: p-value , (should be italic), abbreviations minutes, hours and spelling mistakes (gallic acid/ galic acid) are quite frequent and should be corrected.

>>Response: I have corrected it.

Comments of Editor 1:

Comment 5). Table 3 and figure 8 S.Aureus should be corrected.

>>Response: I have done it.

Responses of Editor Comments

Comments of Editor 1:

Comment 6). Table 7 replace comma with a dot.

>>Response: I have done it

Comments of Editor 1:

Comment 7). Why supplementary figure S4, 5 and 6 with 2D and 3D images are represented separately?

>>Response: We represented Supplementary Figures S3, S4, S5, and S6 separately to provide both 2D and 3D visualizations of each protein–ligand complex. Since our study involves four different proteins with their respective ligands, presenting the 3D docking poses alongside the 2D interaction diagrams allows a clearer understanding of binding orientation, key interacting residues, and molecular contacts. This dual representation ensures a more comprehensive depiction of the binding interactions that could not be fully conveyed by a single format alone.

Comments of Editor 1:

Comment 8). The text fonts should be same throughout the article including figure and graphs.

>>Response: I have done it.

Comments of Editor 1:

Comment 9). As mentioned earlier, kindly go through the article to improve the sentences (section 2.1.4.3, 2.25.2, ) and quality of article so that it can meet the publishing quality of the journal.

>>Response: I have changed and rewritten according to your instructions.

---

## [Editor Report · Decision Letter 5]

13 Oct 2025

Green Synthesis of Silver Nanoparticles from Brownlowia tersa Leaf Extract: Multifaceted Evaluation of Antibacterial, Antioxidant, Cytotoxic, and Anti-Alzheimer Potential

PONE-D-25-15530R5

Dear Dr. Nizam,

We’re pleased to inform you that your manuscript has been judged scientifically suitable for publication and will be formally accepted for publication once it meets all outstanding technical requirements.

Kind regards,

Sana Shamim

Academic Editor

PLOS ONE

Additional Editor Comments (optional):

The article is responded well but still need some improvement regrading Fig 3 (label metal peak) and linguitsics. 
---

## [Editor Report · Acceptance letter]

PONE-D-25-15530R5

PLOS ONE

Dear Dr. Hasan (Ph.D),

I'm pleased to inform you that your manuscript has been deemed suitable for publication in PLOS ONE. Congratulations! Your manuscript is now being handed over to our production team.

Kind regards,

on behalf of

Dr. Sana Shamim

Academic Editor

PLOS ONE